# Chimpanzees adapt their exploration to key properties of the environment

Lou M. Haux [1] ✉, Jan M. Engelmann[2], Esther Herrmann [3,4] & Ralph Hertwig [1,4]

Exploration is an important strategy for reducing the uncertainty that pervades daily life. Yet the evolutionary roots of adaptive exploration are poorly understood. We harness and adapt the human decisions-from-experience paradigm to investigate exploration under uncertainty in chimpanzees. In our study, chimpanzees ($N = 15$; eight females) are simultaneously confronted with an uncertain option (with outcome variance) and a safe option (without outcome variance) and tested in both stable and changing environments. Results reveal that, as in human exploration, how and how much chimpanzees explore depends on the environment. One key environmental property is change: Chimpanzees explore more across trials in changing than in stable conditions. Consistent with the assumption of classic economic models that variance indicates risk, chimpanzees also explore more when they experience variance in the options' outcomes. Individual risk and uncertainty preferences did not have a statistically significant effect on exploratory efforts. These findings suggest that chimpanzees and humans share key similarities in the way they respond to risk and uncertainty.

The core function of cognition is to enable organisms to deal effectively with environmental complexity and the uncertainty it brings[1]. One way of coping with an uncertain world is exploration: gathering information and thereby reducing uncertainty. The investigation of exploratory strategies preceding choices with tangible consequences offers a window onto organisms' ability to reckon with uncertainty[2,3]. Explorative behavior has been found to depend on properties of both the environment and the organism[4,5]. Humans engage in more exploration in unpredictable and complex environments[6–8] and adapt their exploration strategies to the environment[9]. Specific dispositions that vary across individuals, such as openness to experience and risk preference, have also been associated with explorative behavior[10,11]. Yet the evolutionary roots of humans' adaptive exploration remain poorly understood. Investigating the exploration strategies of chimpanzees (*Pan troglodytes*) promises to shed light on how adaptively humans' closest living relatives seek information about the options they face.

Previous behavioral studies investigating strategic information search in nonhuman primates have focused almost exclusively on whether chimpanzees searched for a desired object rather than on how environmental and individual properties influence search[12–16]. In psychology, economics, and neuroscience, information has traditionally been classified as instrumental or non-instrumental. It is considered instrumental if it helps the organism, generally speaking, to achieve a particular end —for example, to alter the course of events or to optimize reward outcomes (e.g.[17,18]). Non-instrumental information has no such value because it cannot serve such purposes. Research on non-instrumental information seeking in humans and other animals has flourished in recent years. The studies[19–23] found that chimpanzees seek information even when it has no instrumental value, indicating that they draw utility from the information itself or from cognitive states triggered by the information[24,25]. This interpretation challenges traditional utility-based models in

[1]Center for Adaptive Rationality, Max Planck Institute for Human Development, Berlin, Germany. [2]Department of Psychology, University of California, Berkeley, Berkeley, CA, USA. [3]School of Psychology, Sport and Health Sciences, University of Portsmouth, Portsmouth, UK. [4]These authors contributed equally: Esther Herrmann, Ralph Hertwig. ✉e-mail: haux@mpib-berlin.mpg.de

economics, according to which utility is reserved for the material outcomes of a choice or action, and information serves only to support the attainment of goals.

We adapt the decisions-from-experience paradigm established in research on risky choice in humans to investigate how chimpanzees explore uncertain environments[2]. In our experimental set-up, chimpanzees chose between two initially unknown options. One was safe, always offering the same outcome, and one was uncertain, offering variable outcomes (including nothing). Chimpanzees could explore these options without costs (i.e., there was no exploration–exploitation trade-off[26]) and then made informed choices based on their exploration. It was up to the chimpanzees to decide how and how much they explored[2]. In our study, the information that the chimpanzees were able to acquire is considered instrumental: By exploring the options, they could learn about the attributes of the options and thus make an informed choice about their preferred option. The long-term expected value of both options was identical, but chimpanzees were unaware of this. Furthermore, exploration allowed the chimpanzees to choose according to their risk preference; in the economic definition, a higher risk preference is reflected in a greater willingness to engage in choices with higher outcome variance (see Fig. 1)[2,5]. Once exploration was terminated by the experimenter, chimpanzees could choose between the two options. Note that some degree of uncertainty persisted even after extensive exploration due to the stochastic nature of the options[2] (see Proof of Concept in SI). Adapting the decisions-from-experience paradigm for use in chimpanzees thus enabled us to investigate whether and how chimpanzees' explorative behavior under uncertainty is shaped by environmental properties and individual risk and uncertainty preferences.

## Results

### Effect of environmental change

We investigated the effect of environmental change on exploratory behavior by exposing chimpanzees to stable and changing environments. The model estimate for the interaction between condition and trials was negative ($b = -0.21$ [$-0.33$, $-0.09$]), suggesting that, over trials, chimpanzees explored changing environments more than stable environments (Fig. 2A; see RQ1 and Table S2 in the SI). The uncertainty around the estimates is likely due to marked interindividual differences in exploration (Fig. 2B).

### Effect of outcome variance

We studied whether chimpanzees were more likely to open trays in the uncertain than in the safe option, conditioned on them actually experiencing outcome variance in the former. Across environments, chimpanzees were significantly more likely to explore the uncertain option when they had experienced outcome variance than when not ($b = 3.47$ [3.13, 3.82]; Fig. 2C; see RQ2 and Table S3 in the SI).

### Exploration strategies and switching behaviors

At least three distinct strategies can be used to explore a set of two options: exploring exclusively only one option; extensively sampling from one option before switching to the other (*sequential exploration*); or switching back and forth between options (*piecewise exploration*)[27]. Across individuals and conditions, chimpanzees limited their exploration to just one option in half of the trials (*Mdn* = 0.48) and explored sequentially in one third of trials (*Mdn* = 0.35) (Fig. 2D). In the changing environment, chimpanzees explored just one option in about half of the trials (*Mdn* = 0.57); and explored

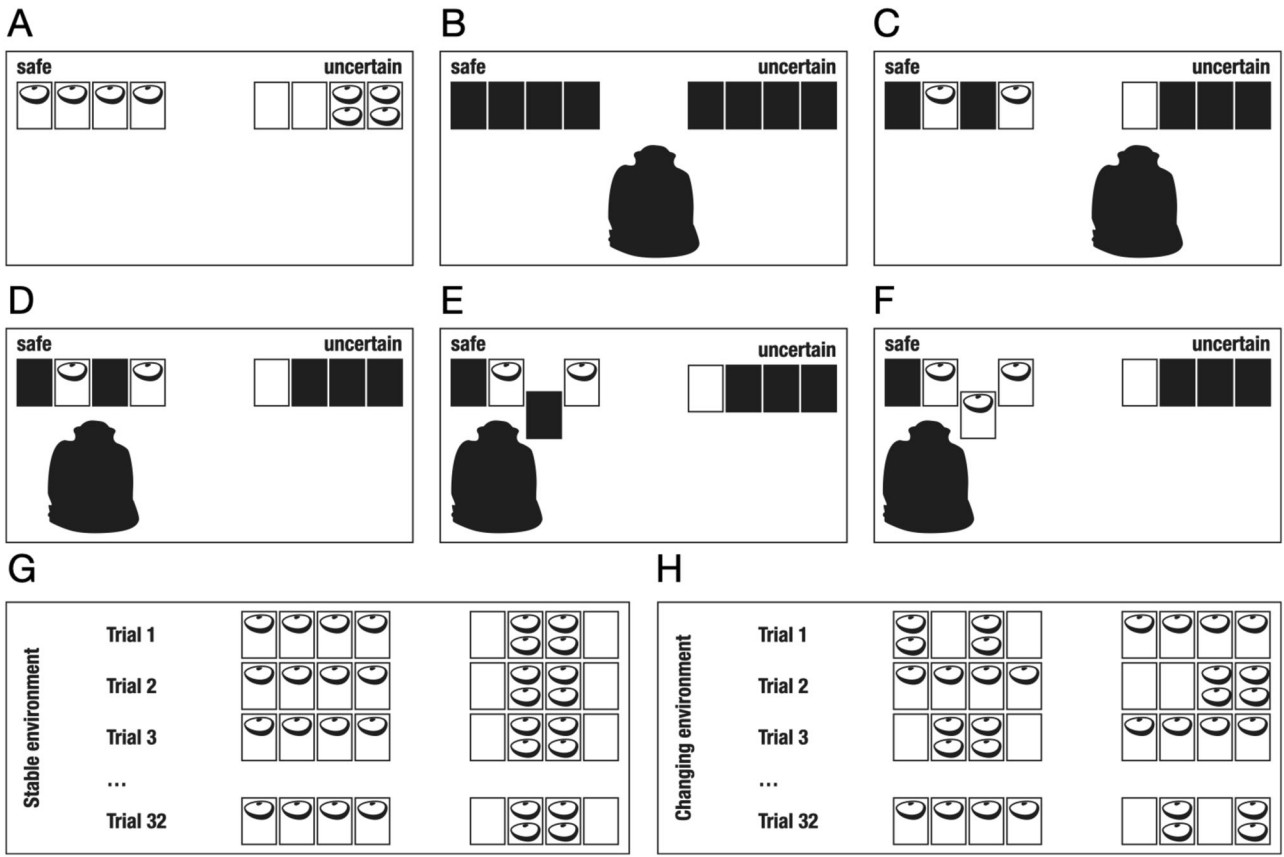

**Fig. 1 | Schematic overview of the experimental set-up. A** Underlying payoff structure. **B** Presentation of two options. **C** Exploration. **D** Choice of one option. **E** Random draw. **F** Reward (or no reward). **G** Stable environment. **H** Changing environment.

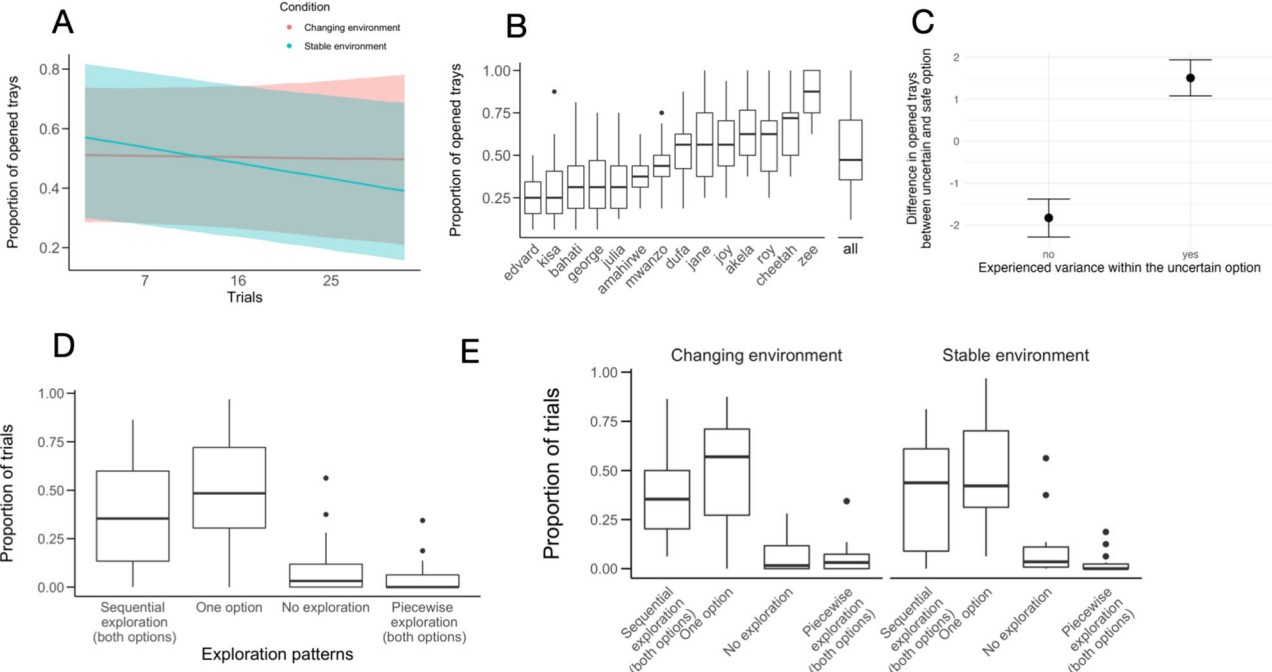

**Fig. 2 | Adaptive exploration.** Plots are based on all available trials for each chimpanzee. **A** Effect of condition on exploration effort. Lines represent regression splines depicting mean levels of opened trays for changing environments and stable environments, with shaded areas indicating 95% credible intervals. Red indicates a changing environment, and blue indicates a stable environment. **B** Interindividual differences in exploration effort across conditions. **C** Mean difference in the number of opened trays between the uncertain and the safe option. A positive difference indicates more exploration in the uncertain than in the safe option. Black dots represent means; error bars represent 95% credible intervals. **D** Exploration strategies and their proportional use across trials. **E** Condition-specific exploration strategies and their proportional use across trials. **B**–**E** Box plots show the median and 25th and 75th percentiles; whiskers indicate the values within 1.5 times the interquartile range; circles are outliers.

sequentially in one third of trials ($Mdn = 0.35$). In the stable environment, chimpanzees focused on one option ($Mdn = 0.42$) and explored sequentially ($Mdn = 0.44$) in about half of the trials (Fig. 2E; see RQ3 in SI).

### Exploration effort

On average, chimpanzees explored only half of the trays, albeit with high interindividual variation. This mirrors findings in humans, which suggest that humans typically explore little and instead rely on surprisingly small samples[5]. Given that exploration is commonly associated with opportunity and processing costs, limiting exploration effort can be rational[28] and may even be advantageous.

### Risk and uncertainty preferences

To examine the impact of individual dispositions, we investigated the relationship between mean exploration effort (number of opened trays) and chimpanzees' risk and uncertainty preferences, as measured in a previous study[29]. In psychology, risk preference describes the tendency to exhibit behaviors that, though rewarding, involve potential harms or losses; risk preference is often measured by stated preferences. In economics, preferences are assessed through behavioral choice experiments in which the potential outcomes and their probabilities are either known (risk preference) or unknown (uncertainty preference). Our analyses use measures from both disciplines. Risk preferences were measured in two ways: (A) by means of behavioral choice experiments in which the potential outcomes and their probabilities were known; (B) by means of observer reports in which longtime caregivers rated chimpanzees' willingness to take risks in general and in five domains representing major classes of risks in chimpanzees' ecology. Uncertainty preferences were measured using behavioral choice experiments in which the potential

outcomes and their probabilities were unknown. Risky and uncertain choice behavior was previously found to be negatively correlated ($-0.50$ [$-0.70$, $-0.23$])[29]. Here, we calculated Pearson correlations and found no statistically significant correlation t(11) = $-0.34$, p = 0.74, r = $-0.10$, 95% CI = [$-0.62, 0.48$] between chimpanzees' mean exploration effort and the behavioral risk measure and no statistically significant correlation t(12) = $-1.70$, p = 0.12, r = $-0.44$, 95% CI = [$-0.79, 0.12$] between chimpanzees' mean exploration effort and the observational risk measure for general risk. Furthermore, we found no statistically significant correlation t(11) = 1.13, p = 0.28, r = 0.32, 95% CI = [$-0.28, 0.74$] between mean exploration effort and the behavioral uncertainty measure.

### Discussion

Our results suggest that chimpanzees, like humans, flexibly adapt their exploration to key properties of the environment. First, change in the environment seemed to attract chimpanzees' attention and affected their explorative behavior: Chimpanzees explored more across trials in changing environments than in stable ones. Relatedly, in changing environments, chimpanzees preferred to explore just one option, whereas in stable environments, they both focused on one option and explored sequentially. This pattern of results is in line with previous findings suggesting that, if expectations about an option are violated, explorative behavior will be directed toward this option[7]. Moreover, focusing on one option reduces energy and memory costs[27]. Second, the experience of outcome variance increased investment in exploration: Chimpanzees were more likely to open trays in the uncertain option than in the safe option when they had experienced outcome variance in the former. These results are in line with earlier findings suggesting that humans explore riskier options more than safer options when they experience outcome variance—a reasonable response to the signal of risk[9].

Chimpanzees' risk and uncertainty preferences did not have a statistically significant effect on their explorative behavior. Theoretical models suggest that risk and uncertainty preferences are independent, which would imply that a risk-averse individual could be uncertainty-neutral, uncertainty-loving, or uncertainty-averse. Empirical results in humans are mixed, with some studies finding a positive or negative correlation and other studies detecting no significant correlation[30,31].

Taken together, our results indicate that humans and nonhuman primates share notable similarities in exploration—a key cognitive tool for reducing uncertainty in the environment. Both chimpanzees and humans appear to tailor their exploration efforts to properties of the choice environment. These findings contribute another piece to understanding the evolution of decision-making processes.

## Methods

### Participants

We tested 15 semi-free-ranging chimpanzees from Sweetwaters Chimpanzee Sanctuary in Kenya (eight females; age: $M = 22.83$ years, range = 13–33 years; for individual characteristics, see Source Data S01).

### Ethics and Inclusion

Understanding chimpanzee behavior can develop more effective strategies to conserve and protect this endangered species in the sanctuaries and in the wild. Roles and responsibilities were agreed amongst collaborators ahead of the research. We are grateful to Richard Vigne, Samuel Mutisya, Stephen Ngulu, the board members and staff of Sweetwaters Chimpanzee Sanctuary in Kenya for their support (in particular rendering possible access to the chimpanzees during data collection). The current research has been approved by Ol Pejeta Conservancy, Kenya Wildlife Service (KWS), and the National Council for Science and Technology (NCST).

### Animal welfare

The research was noninvasive and carried out in accordance with the guidelines of the Pan African Sanctuary Alliance and the regulations of Sweetwaters Chimpanzee Sanctuary, Ol Pejeta Conservancy, in Kenya. The full procedure of the study was approved by the local ethics committee at the sanctuaries (board members and veterinarian), Kenya Wildlife Service, and the Kenyan National Council for Science and Technology. Chimpanzees at the Sanctuary have access to large tracts of outdoor enclosures, including trees, bushes, and climbing structures, and live in large, mixed-sex social groups. They are fed a combination of fruits, vegetables, and other species-appropriate foods three times daily. All individuals stay in indoor enclosures overnight. Chimpanzees were tested in familiar rooms and never deprived of food or water for any reason. All testing was strictly voluntary. A chimpanzee could stop participating at any time by, for instance, heading to the door or not making a choice. All chimpanzees were highly motivated to participate. Due to the COVID-19 pandemic, 5% of planned trials could not be run (see Source Data S01 for details on missing trials). We excluded the data of one chimpanzee, Niyonkuru, as he was tested in only one condition.

### Study design

Figure 1 provides a schematic view of the experimental set-up. Chimpanzees chose between a safe option (where the outcome did not vary; i.e., without outcome variance within the option) and an uncertain option (where the outcome varied; i.e., with outcome variance within the option; Fig. 1A). Each option consisted of four covered trays. In the safe option, each tray was baited with a quarter of an apple. In the uncertain option, two trays were baited with half an apple (two quarter pieces) each and two trays were empty. The expected value over several trials was thus the same for both options. Chimpanzees began with

no knowledge of the payoff distributions and could learn about the possible outcomes and their frequencies by drawing random samples from each option (Fig. 1B–F; Supplementary Movie 1). This exploration process was fully under the chimpanzees' own control: They could decide whether to explore, which option to explore, and when to switch between options. During the exploration phase, chimpanzees could not retrieve any of the options; they had one minute to explore the trays' content by pulling the lid ropes to open the corresponding lids. After the exploration phase, the experimenter removed the lid ropes and made the choice ropes available. Chimpanzees chose one of the options by pulling the corresponding rope. One tray was then randomly drawn from the chosen option (safe or uncertain), and the chimpanzee obtained that reward (or no reward if the tray drawn was empty).

In a within-subjects design, chimpanzees participated in both a stable and a changing environment condition. Each condition comprised 32 trials, presented across eight sessions of four trials. The conditions were blocked, meaning that chimpanzees first completed all trials of one condition, followed by the other condition (the order of conditions was counterbalanced across chimpanzees). In the stable environment (Fig. 1G), the safe and uncertain options each stayed on the same side over trials and the same trays in the uncertain option were baited with food. In the changing environment (Fig. 1H), the safe and uncertain options changed sides over trials, and the position of the baited trays in the uncertain option varied.

### Statistical analysis

We employed Bayesian estimation techniques. Specifically, we conducted regression analyses using Bayesian generalized linear models implemented in R[32] using the Stan software, with the brm function from the brms package[33]. We specified weakly informative normal priors with mean 0 and standard deviation 2 on all population-level effects[34]. We assessed the convergence of posteriors through visual inspection and the Gelman–Rubin diagnostic, Rhat, with a cut-off value of 1.01[35]. For the regression analyses using Bayesian generalized linear models, we report the mean of the posterior distribution of the parameter and two-sided 95% equal-tailed credible intervals (CI) around each value. We further computed Bayes factors to test whether the parameter differed from zero, using the hypothesis() function in the brms package ([36], see SI). For all data figures, we used the function conditional_effects to display the conditional effects of the predictors of the fitted models. We computed leave-one-out cross-validation (LOO) values for every model ([37], see SI). The LOO value indicates a model's pointwise out-of-sample prediction accuracy; models with higher LOO values are preferred. For the model comparison, we also added LOO weights (weights add up to 1). We used these metrics to rank models[38]. For the relationship between mean exploration effort (number of opened trays) and chimpanzees' risk and uncertainty preferences, we report correlations with confidence intervals[39].

### Reporting summary

Further information on research design is available in the Nature Portfolio Reporting Summary linked to this article.

## Data availability

All data associated with this manuscript are available on GitHub: https://doi.org/10.5281/zenodo.13907943. Source data are provided with this paper.

## Code availability

R scripts associated with this manuscript are available on GitHub. Exact computational reproducibility might only be achievable by using the model fit objects: https://doi.org/10.5281/zenodo.13907943.

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

## Acknowledgements

We thank Richard Vigne, Samuel Mutisya, Stephen Ngulu, the board members and staff of Sweetwaters Chimpanzee Sanctuary in Kenya for their crucial support during all stages of this research. We thank Ol Pejeta Conservancy, Kenya Wildlife Service (KWS), and the National Council for Science and Technology (NCST) for approving our research. Thanks go to Larissa Samaan for reliability coding, Susannah Goss and Deb Ain for editing the manuscript, Sarah Otterstetter for help with the figures, Dominik Deffner for methods advice, and Tomás Lejarraga, Leonidas Spiliopoulos, and Dirk Wulff for helpful discussions.

## Author contributions

L.H., J.M.E., E.H. and R.H. conceived of the project. L.M.H. conducted the research and analyzed the data. L.M.H. wrote the paper with input from J.M.E., E.H. and R.H. E.H. and R.H. contributed equally.

## Funding

## Competing interests

The authors declare no competing interests.
