## [Transparent Peer Review File · Nature Communications]

Chimpanzees adapt their exploration to key properties of the environment

Corresponding Author: Ms Lou Haux

Version 0:

Reviewer comments:

Reviewer #1

(Remarks to the Author)

This experiment examined exploration/information seeking in chimpanzees. Using a task that was adapted from a similar task to study exploration in humans, chimps could preview the contents of a series of four trays at each of two options—a safe option which provided $\frac{1}{4}$ apple in the 4 trays and an uncertain option which provided $\frac{1}{2}$ apple in two trays and no apple in two trays. Following the opportunity to potentially explore all 4 trays at both options, the chimps chose between the two options and were given a random draw of one of the trays from the chosen option.

The noteworthy results were that, as with humans on the original task, the exploration strategies used and how much exploration occurred varied with how the options were presented. For example, exploration increased under changing conditions in which the locations (left vs right) of safe and uncertain locations varied unpredictably across trials. Further, risk and uncertainty preferences of these chimpanzees as assessed in a prior study using other means were predictive of exploration tendencies in the present task.

Little work has been conducted on exploration in chimpanzees. As a result, the reported findings are likely to have a meaningful impact on our understanding of chimp behavior and its similarity to human behavior.

Although the extension of the study of exploratory behavior to chimps is novel, the paper could benefit from a more in-depth consideration of information seeking in the Introduction. There is only a cursory mention of other work on information seeking in non-humans. In addition to the other work cited, there is a burgeoning literature on “non-instrumental information seeking” in neuroscience (some of which has been conducted with other primates and some with humans; example references below). This literature has ties to a much larger and older literature on “observing response procedures” as a way to study how environmental properties like uncertainty affect information seeking across species from fish to humans. As with the information provided by exploration in the present task, non-instrumental information (and the information provided by observing responses) cannot be used to change event outcomes. Although the information provided to the chimps could be used to choose one of the options, choosing one option over the other had no meaningful impact over the long run (they would receive on average $\frac{1}{4}$ apple per trial over the long run). The present paper should properly place its examination of information seeking in chimps into this long running, yet still current, literature.

Bennett D, Bode S, Brydevall M, Warren H, Murawski C (2016) Intrinsic Valuation of Information in Decision Making under Uncertainty. *PLOS Computational Biology* 12(7): e1005020. <https://doi.org/10.1371/journal.pcbi.1005020>

Bromberg-Martin ES, Monosov IE. Neural circuitry of information seeking. *Curr Opin Behav Sci.* 2020 Oct;35:62-70. doi: 10.1016/j.cobeha.2020.07.006.

Daddaoua, N., Lopes, & Gottlieb, J. Intrinsically motivated oculomotor exploration guided by uncertainty reduction and conditioned reinforcement in non-human primates. *Sci Rep* 6, 20202 (2016). <https://doi.org/10.1038/srep20202>

Overall, the methods used are appropriate and the data support the authors' claims. However, in both cases the descriptions in the main body of the text are very unclear and lacking in critical details. The information is available in the supporting information, but some of that material is essential for even a basic understanding of the study and needs to be moved to the main body of the paper. This is particularly true of the basic description of the task. The two paragraph description of the task under “Test Phase” on page 20 in the supporting information should be moved to the body of the text. Without it, the procedure is incomprehensible, even with the figure.

Finally, the finding that chimps that were more tolerant of uncertainty searched more rather than less (the opposite of what was found with humans) needs far more discussion. This finding is mentioned, but no potential explanations are explored. This finding is very difficult to understand in the text of the other findings and could raise concerns about the detection of potentially spurious relations in the correlational data. It needs to be addressed fully in the Discussion section

Timothy Shahan

Reviewer #2

(Remarks to the Author)

This is a very interesting and innovative paper exploring an understudied topic in nonhuman cognition. Ideas about how exploratory behavior relate to risk and environmental stability are important for understanding the evolution of cognition more broadly, as well as for exploring similarities and differences between humans and their closest relatives. The manuscript is generally well written and the experiments appear to have been rigorously conducted. I do not have any major concerns with the methodology. My concerns center mostly around the implications of the work, which are not fully explored, and the statistical approach.

It is difficult to understand the outcomes on p. 5 without first knowing the expectations for the chimpanzees. Were they able to retrieve as many food items as possible or were there limitations? If not, then why would they not just open all options? What would be the motivation to explore only a single option? I realize the main text has to be very brief but at least some more detail of what the task looks like for the chimps is needed for the reader to understand. Is there a choice to be made after some period of exploration? This is clear in the SI on p. 5 but not in the briefer main text.

I need to know more about how the authors are distinguishing risk-seeking from uncertainty-tolerating before the findings at the bottom of p. 6 can make sense to me. At first, it seems counter-intuitive that risk-seeking chimpanzees explore less without the framing presented in the abstract that they are willing to take on greater uncertainty. This could be spelled out more explicitly in the main text as well. More detail about the findings of the previous study are warranted.

I don't think the random draw concept that appears in Figure 1 was discussed in the text. My preference would be for this kind of study to be presented in a longer more typical format as it seems the supplementary text is fundamental to understand the procedure.

I confess that I am not familiar with running Bayesian statistics so I cannot comment on the appropriateness of the statistics in this section. It seems like the authors are interpreting confidence intervals containing 0 as significant, however, when this normally indicates lack of significance (p. 4 of SI).

Why was there no criterion in the third pre-test? Didn't the chimpanzees have to demonstrate that they understood they could pull the lid ropes to explore the options?

Please define choice clearly. Was a chimpanzee considered to have made a choice as soon as they touched one of the ropes or as soon as they pulled it at all?

The authors should conduct simple slopes tests to explore interactions of the regressions (p. 9). Given that they predictors are discrete and nested rather than continuous (except for trial), why use a regression rather than a generalized linear model with binomial distribution? Similarly, for accounting for whether they experienced outcome uncertainty in a given trial, could you not use a GLM?

Many analyses were conducted on the same outcomes. The authors should employ some sort of correction.

Is it possible to distinguish between interpretations based on recognition of safe and uncertain options versus responses to empty trays (only present in the uncertain option)? Similarly, how can we be certain that chimpanzees recognized the difference between stable and unstable conditions? I recognize that they behave differently in the two conditions and that a pilot study was done, but chimpanzees are so impulsive – at what point in the session do they recognize which condition is operating?

More needs to be said about the implications for understanding the evolution of decision making or at least for cross-species comparisons. Again, I feel that the short format here does not do this study justice.

How exactly did the experimenter pull the lids back safely and how did they present the choice ropes to the chimpanzees – simultaneously or one at a time? If so, in what order? How was the chimpanzee prevented from pulling a rope before both were set up? More details as to how the trial was run are needed.

I'm intrigued by the exploration of win-stay versus win-shift. I am surprised a chimpanzee would not simply choose a side where they have seen the most food, or any food. I feel like this analysis also needs more space for discussion.

I'm a bit confused by whether the same eight chimpanzees participated in both experiments plus an additional seven chimpanzees for the main experiment?

The authors need to insert more commas between clauses.

Reviewer #3

(Remarks to the Author)

In this article, the authors investigate risk preferences in stable and changing environments in chimpanzees. They use a four-alternative choice paradigm with safe and uncertain options, where safe options are guaranteed to deliver a smaller reward and uncertain options can deliver a large reward or no reward. The expected values across all options in both conditions are the same. Chimps could explore options before choosing. In the stable environment, which set of options were safe and which uncertain did not change, whereas in the changing environment, the safe and uncertain options could

switch. The authors claim to have discovered that chimps boost exploration for uncertain outcomes, adapt their exploration studies to environmental stability, and to have uncovered a correlation between individual preferences and exploration.

This manuscript is poorly written, confusing, poorly explained, and the conclusions are not supported by the findings (which are unclear to begin with). There is a second entire manuscript in the supporting information. Many of the critical details of the experiment are in the supporting information and need to be re-written and moved to the manuscript, but many more analyses are left out altogether, including entropy analyses that are completely absent from the paper. I advise the authors to completely re-write the manuscript.

Score: 2/10

Recommendation: Reject

The paper has SO many problems.

1. It is very poorly written and difficult to understand. Variables are not explained or presented; in-text methods summaries are absent; and clear definitions are absent.
 2. The scientific value of the manuscript is unclear. The authors cite very little literature. This is a shame and makes the manuscript challenging to evaluate. There is a lively research program looking into information search in non-human primates but the authors cite almost none of it.
 3. The supplement contains an entire manuscript's worth of detail on another experiment, as well as many details that need to go into the manuscript.
- I could go on but I think the message is communicated.

Introduction:

1. This paper evinces shockingly poor scholarship. There is a ton of work looking at information search in NHPs: Barack et al. 2023; reviews in Wilson et al. 2022 and Bromberg-Martin 2020; look at information search in Gottlieb's work; Blanchard et al. looking at observing responses in macaques; even basic neuroeconomic studies investigating risk preferences. There is so much work investigating information search, the impact of uncertainty on choice, and related topics, and the authors blithely ignore this work.
2. In part because the authors don't situate their work in context, it is unclear how to situate their findings or even properly understand them.

Methods:

1. The "Study Design" was far too short and unclear to know what the chimps actually did.
2. Some questions:
 - o There was no quantification of outcome entropy or entropy reduction across the changing environment. How do these entropy measures impact choices?
 - o The use of a switch between stable and changing environments invokes additional cognitive change detection processes that must be modeled. See Josh Gold's work on change detection. How did change detection regulate choice?

Results:

1. The results were unclear and inappropriately reported.
2. Example 1: "Comparisons between binomial regression models (see SI) showed that those including the test predictors condition and trial made better predictions, with the model including the interaction between condition and trials performing best": The 'condition' and 'trial' covariates were never defined in the main text.
3. Example 2: "Mean exploration effort and all risk measures were negatively correlated (behavioral risk measure: $-.10 [-.62, .48]$; observational risk measure, e.g., general risk: $-.44 [-.79, .12]$;...), meaning that risk-seeking chimpanzees explored less. Furthermore, we found positive correlations between mean exploration effort and the behavioral uncertainty measure ($.32 [-.28, .74]$), suggesting that uncertainty-tolerant chimpanzees explored more": as far as I can tell, none of these measures are significant because their confidence intervals overlap with 0, but given the complete lack of supporting information about these analyses, it is totally unclear.
4. There was no quantification of uncertain (aleatory or epistemic) e.g. using entropy measures. This is unforgivable.
5. It was unclear if fine-grained measures of information gathering (e.g., number of opened trays by trial; timing of decision to open trays during the exploration of phase; etc.) were included in regressions.

Figures:

1. This figure needs to be completely re-done; instead of placing each phase in a trial in a different panel, the phases should be in the same panel and labeled appropriately.
2. 2A and 2C seems to show no effect. D and E collapse across important information about how the animals explore.

Discussion:

1. "change in environments attracted attention and affected explorative behavior": what does attention have to do with this?
2. "as defined in the economic definition of risk": never defined this.
3. "We observed marked interindividual differences in exploration, with risk-seeking chimpanzees tending to explore less. Notably, uncertainty-tolerating chimpanzees tended to search more rather than less": unclear what analyses supported this conclusion.

Reviewer #4

(Remarks to the Author)

Summary:

The present manuscript describes a behavioral study in chimpanzees investigating exploration and choice behavior in stable and changing environments.

Review:

I was asked by the handling editor to review the methodological and statistical aspects of the manuscript, so I will focus on these issues in my review. Note that I am not an expert in comparative cognition, so I cannot evaluate the theoretical contribution of the project.

Overall, I enjoyed reading the manuscript and I thought that the study was summarized clearly and concisely. I also applaud the authors for making the data and the video recording of the experimental design available. With regard to the description of the study design and the results, I think the manuscript could benefit from some clarifications that should be incorporated into a revised version of the manuscript.

- (1) When reporting and discussing the results, I think that more consideration could be given to an adequate representation of uncertainty. From a statistical perspective, the sample size is relatively small, which leads to large credible intervals for reported between-subject effects. For example, this becomes particularly obvious in the correlations between exploration effort and personality characteristics where plausible correlation coefficients range between large negative and medium-sized positive correlations (e.g., hierarchy risk: $-.62$ to $.48$). In my view, this uncertainty is not adequately represented in the verbal reporting of results where only the direction of the point estimate is mentioned. Moreover, I think the wider generalizability from a sample of 15 chimpanzees to a larger population should be discussed.
- (2) To be able to check and fully reproduce the presented statistical analyses, it would be very helpful if the analysis code was made available and there was a codebook for the datasets that describes the variables.
- (3) In general, it would be good if full result tables with model coefficients could be presented somewhere for the binomial regression models. Since these models are fairly complex, they have more parameters than what is reported in the text (i.e., main and interaction effects), and I think it would be interesting at least for part of the readers to see the full results. This does not have to happen in the main text if this makes the manuscript too long, but perhaps this could go in supplementary online materials. Potentially, it could even be resolved together with my previous point in the form of a commented analysis file (e.g., RMarkdown, Quarto, etc.) that renders the analysis code and output tables together. I'll leave this to the authors to decide.
- (4) On p. 3 (SI), the manuscript says: "The LOO value indicates a model's pointwise out-of-sample prediction accuracy; models with lower LOO values are preferred". Actually, models with higher LOO values (i.e., less negative values) should be preferred (see e.g., <https://vasishth.github.io/bayescogsci/book/ch-cv.html>; https://mc-stan.org/loo/reference/loo_compare). I think this is just a minor mistake that does not influence the interpretation of results because later in the manuscript models are correctly selected with for higher LOO-weights. However, the authors should probably check this.
- (5) I was a bit confused as to what exactly the dependent variable in the binomial regression models was. Since the models were called "binomial regression" rather than "binomial logistic regression", I initially assumed that it would be the number of explored trays (0 to 4 out of 4 in each option). However, based on the reporting in the manuscript, it sounds like the result is dichotomized (exploration yes/no). It would be good if the authors could elaborate more on the choice of the dependent variable, specifically, give a reason for the dichotomization (if the variable was actually dichotomized). Alternatively, the analysis could be changed to model the raw data, which should also increase statistical power.
- (6) In the description of the binomial regression modeling results, to get full information on the fitting procedure, it would be good to report the cut-off value used on the Gelman-Rubin statistic (best practice would be 1.01), the total number of samples and chains, as well as the bulk and tail effective sample sizes.
- (7) In the SI, it is mentioned in several places that "condition was dummy-coded and centered". I am not sure if this makes sense. If a binary variable (condition) is dummy-coded, levels are identified as 0 and 1. Centering means subtracting each value from the mean, i.e., subtracting 0 or 1 from a value between 0 and 1, yielding a positive and a negative value that are not equidistant from zero unless the mean is exactly 0.5 (which it is if there are equally many instances of condition 0 and 1, but then the coding is equivalent to effect coding, and it would be easier to talk about effect coding in the first place). It would be good to add some explanation here. Also, as a minor point on coding, it would be good to mention somewhere explicitly which condition was used as a baseline in dummy-coding, so that this doesn't have to be inferred indirectly from the parameter interpretation.
- (8) There are several instances of "hidden" hypothesis tests in the manuscript where the authors use the information whether a credible interval covers zero as evidence for whether the parameter is zero or not. A more principled approach would be to compute and interpret a Bayes factor to test whether the parameter differs from zero. This can be easily done using the hypothesis() function in the brms package that the authors already employ (<https://paul-buerkner.github.io/brms/reference/hypothesis.html>). For a more detailed explanation for why credible intervals should not be used for hypothesis testing, see e.g., <https://osf.io/preprints/psyarxiv/rqnu5>.
- (9) To me, it is unclear why first a "best" model is selected using the LOO cross-validation method, but then main effect estimates are not interpreted in the context of the winning (condition x trial) model, but in the context of a simpler model (main effects model excluding the interaction, see p. 4 SI). Maybe this is only a misunderstanding on my part, but since the condition x trial model should also include main effects, I think these should be interpreted in the context of the winning, more complex, model.
- (10) There are a few switches between Bayesian and frequentist analyses methods in the manuscript that I don't understand conceptually. For example, in Figure 2C, according to the caption, confidence intervals are reported instead of credible intervals. Moreover, p-values are reported for Wilcoxon-signed-rank tests instead of Bayes factors or LOO-CV coefficients. I don't think that these days anyone should have to justify why they use one framework or the other, but to me, it seems a bit

unprincipled to switch between statistical frameworks without a substantive reason. For example, Bayesian Wilcoxon signed-rank tests could be easily computed with readily available packages in R (see e.g., <https://forum.cogsci.nl/discussion/8322/bayesian-wilcoxon-signed-rank-test-in-r> for two resources).

Minor points:

- I got a bit confused at some point because different phrases were used to describe the bait/reward at different times in the manuscript. As far as I understand, in the main experiment the pieces were always a quarter of an apple, and in the proof-of-concept study, they were half an apple, but throughout the manuscript, sometimes the piece size and sometimes the overall reward size was used which was a bit confusing to me (e.g., p. 5 SI: "In the safe option, each tray was baited with a quarter of an apple. In the uncertain option, two trays were baited with half an apple each", but according to Figure 1, the "half apple" are actually two pieces of a quarter apple and not one bigger half-apple piece. There are several occurrences like this throughout the manuscript.)
- Figure 1H / operationalization of changing environment: It wasn't clear to me if the same pattern was maintained across subjects (e.g., the displayed trial 1 was shown first to all subjects) or if the order was permuted across subjects. This could be made clearer, and in case that there was no permutation, the full pattern could be presented in an appendix.
- In the SI, it would be good to introduce more consistent labeling of sections and sub-sections to make the structure clearly visible. For example, I found it confusing that the Methods section on p. 1 is at the same structure level as the "proof of concept" section before although both describe the proof-of-concept study. For "Experiment", all following sections were introduced as sub-sections, but some of them referred to the familiarization phase which was basically a pre-test with its own experimental setup, so took a bit more time than necessary (for me, again, I'm not familiar with the subject area) to disentangle the experimental setups.
- It would be good if some information was included on how many chimpanzees (if any) had to be excluded based on pre-tests.

Angelika Stefan

Version 1:

Reviewer comments:

Reviewer #1

(Remarks to the Author)

Overall, the revised paper is much improved. The methods are considerably easier to follow, and the discussion of results is much clearer.

One remaining issue is the treatment of previous related work with non-humans (including primates) on observing responses and "non-instrumental" information seeking. In both cases, animals (including humans) work for information that generally does not improve their situation longer term (for example, across trials). In a classic example of observing responses (Prokasy, 1956) rats choose between two options, both of which pay off with $p=.5$. One option always has the same stimulus (or two uncorrelated stimuli) and the other has two stimuli differentially correlated with positive and negative trials. The information is non-instrumental as the animal cannot increase the overall rate of return in the task (see also Dunn et al., 2024 for extension of this to situations where information seeking actually decreases the rate of reward over time). Nevertheless, the animals choose the option with the informative correlated cues. The task the present authors have arranged seems to share many of the same features. The chimps can look into the cups before choosing, but regardless of what they do in terms of information seeking, the overall expected payoff from both options over time is the same, $\frac{1}{4}$ apple. Nevertheless, the authors describe their procedure as being "instrumental" in contrast to observing procedures. I do not think this is correct and I do not see how trying to make the distinction is all that useful here. I realize the journal format does not allow much space, but given the above, I feel like the current treatment of the previous literature still does not work. I think the authors will need to try a different approach to provide a better context for this work. Perhaps it would be enough to briefly describe observing and non-instrumental information seeking, note that the procedure used here (and previously with humans) shares many properties of these long-used procedures (rather than trying to assert that it does something they do not), and then say it is being extended to chimps because little is known about information seeking in chimps.

Timothy Shahan

Dunn RM, Pisklak JM, McDevitt MA, Spetch ML. Suboptimal choice: A review and quantification of the signal for good news (SiGN) model. *Psychol Rev.* 2024 Jan;131(1):58-78. doi: 10.1037/rev0000416. Epub 2023 Mar 6. PMID: 36877476.

Prokasy, W. F., Jr. (1956). The acquisition of observing responses in the absence of differential external reinforcement. *Journal of Comparative and Physiological Psychology*, 49(2), 131–134. <https://doi.org/10.1037/h0046740>

(Remarks on code availability)

Reviewer #2

(Remarks to the Author)

I thank the authors for their careful responses to the reviewers' previous comments. Their explanations have helped to clarify some of the details of the manuscript that were previously unclear to me.

On line 40, do you mean they do not use the information to inform the choice of a better option? That could be even more explicit.

Please clearly define "outcome variance" (line 83). Do you mean within-trial variance in the overall number of rewards that could be obtained?

The blocked design is still not quite clear in the main text (around lines 97-99). Did the chimpanzees receive only a total of eight sessions of each condition for a total of 16 4-trial sessions? Then what order were the conditions presented in? Even in SI, it is not clear if all the trials of one condition were presented before any trials of the other condition.

I would still like to see the traits of risk-seeking and uncertainty-tolerant better differentiated. Please define these terms.

It still isn't clear to me why uncertainty tolerant animals would explore more.

Line 36 "or not" is not needed after "whether."

There are still missing commas after clauses; e.g., after "over trials" on line 124 and after "on average" on line 142. It is especially needed after "in chimpanzees" on line 185. Check throughout.

In the first sentence of the SI Results, "To investigate" is duplicated.

Please use "that" instead of "who" when referring to chimpanzees as is APA style.

(Remarks on code availability)

Reviewer #3

(Remarks to the Author)

The authors did a fine job of responding to withering criticisms.

- There are many studies on instrumental information use in NHPs, but an extra data point is always welcome.

- I find the lack of exploratory behavior in the risk-tolerant participants to be utterly mystifying, but that's not fault of the authors'. Their meager explanation in the discussion ("Risk-seeking chimpanzees tended to explore less, suggesting that they accepted more risks in their environment") is, on one read, an inconsistency but is better read as simply re-describing the finding. I might say something a little more substantial?

- While age was mentioned in the conclusion, it was never mentioned in the results (though reported in the SI tables). I would briefly mention it in the results or remove the mention in the conclusion.

Publish.

(Remarks on code availability)

Reviewer #4

(Remarks to the Author)

I went through the authors' responses, the updated manuscript, and the detailed supplementary materials. I thank the authors for considering the concerns that I previously raised and for their thoughtful responses.

My only remaining concern is that the analysis code will only be shared upon request. I understand that sharing analysis scripts can be daunting, but I would like the authors to consider that it also has many advantages: Analyses become reproducible, the scripts remain accessible even if personal hard drives get lost, and the scripts can be re-used by other researchers for similar analyses. Additionally, writing a request may be perceived as a barrier to accessibility particularly by junior and minority researchers, such that it can be assumed that, practically, the code would be more accessible to some groups of researchers than to others. On the side of the authors, answering multiple requests for sharing analysis code also arguably requires more time and effort than making the code available once. Making code available is also highly recommended by Nature Communications' journal guidelines (<https://www.nature.com/nature-portfolio/editorial-policies/reporting-standards#availability-of-computer-code>). However, I don't want to impose my views on the authors, and I'll leave it to the editor to decide to what extent code availability is necessary for publication.

Angelika Stefan

(Remarks on code availability)

I wasn't able to review the code because it was not shared with me. However, the authors claim that the code is available upon request (see my comments above).

Version 2:

Reviewer comments:

Reviewer #1

(Remarks to the Author)

Overall, the authors have addressed my concerns and I think the paper can be published. Although it is probably still debatable if the information delivered to the chimps is instrumental or not in the context of the modern neuroscientific use of the term, I think the authors should be allowed to characterize it how they like. However, the following sentence (lines 57-58) should be deleted: "Acquiring information was instrumental because search reduced the subjects' prior state of uncertainty." This sentence makes no sense as the very definition of information (both formal and informal) requires uncertainty reduction. If there is no uncertainty reduction, there is no information, so that cannot be what makes it any particular piece of information instrumental or not.

(Remarks on code availability)

The R markdown appears to provide needed details for the analysis and the data appear to be available.

Reviewer #2

(Remarks to the Author)

Thank you for these additional revisions. The paper is much clearer now and I think should be published.

(Remarks on code availability)

Reviewer #4

(Remarks to the Author)

I want to thank the authors for sharing their code in the revised version of the manuscript. I was asked to take a look at the code and provide comments if necessary. To quickly summarize, I think that the authors did an excellent job at making their results reproducible.

I should mention that I haven't had the time to check whether all results that are reported in the manuscript can be reproduced using the code (I also believe that this would be out of the scope of a regular peer review process). However, all code fragments I tried out (approximately $\frac{1}{4}$ of the entire code) ran smoothly. Moreover, for the sample of analyses I reproduced, I also obtained results in line with the ones reported in the manuscript. The code was very well documented and made it easy for me to find the corresponding results in the manuscript. Overall, I would say this bodes very well for overall reproducibility.

In terms of statistical analyses, I took another look at the Bayesian models, and they seem well specified to me. As a minor remark, I think the chosen number of iterations (4000) might be slightly too small in combination with the chosen number of warmup samples (2000) to provide computational reproducibility to the number of digits reported in the manuscript. At least on my machine, I had 1-2 instances where results deviated from what was reported in the second decimal digit due to the algorithmic fluctuations (e.g., an estimate of 0.37 instead of 0.36). I think this point is so minor that I don't think it requires any adjustments in the current code, especially since the results in the manuscript can be reproduced by using the model fit objects that the authors provide on GitHub. However, it might be worthwhile to leave a brief remark in the code or manuscript alerting the reader to the fact that exact computational reproducibility might only be achievable by using the model fit objects.

Another really minor remark in terms of reproducibility would be that I couldn't find information on what versions of the packages were used. This is important to ensure that the analyses can be reproduced in the future. I'd suggest to include this information by including the output of `sessionInfo()` from the computer that was used to run the analyses in the README file on Github.

Signed,
Angelika Stefan

(Remarks on code availability)

see my comments above

RESPONSE TO Reviewer #1:

- 1) *The study investigated exploration tendencies in chimpanzees using a task adapted from human exploration studies. Chimps could preview four trays at two options – a safe one with ¼ apple in each tray, and an uncertain one with ½ apple in two trays and none in the others. After the exploration phase, chimps chose an option, and a tray was randomly drawn from it. The noteworthy results were that, as with humans on the original task, the exploration strategies used and how much exploration occurred varied with how the options were presented. The chimps' risk and uncertainty preferences, assessed previously, predicted exploration tendencies in this task. Little work has been conducted on exploration in chimpanzees. As a result, the reported findings are likely to have a meaningful impact on our understanding of chimp behavior and its similarity to human behavior.*

We thank Reviewer 1 for this positive assessment of our work.

- 2) *Although the extension of the study of exploratory behavior to chimps is novel, the paper could benefit from a more in-depth consideration of information seeking in the Introduction. There is only a cursory mention of other work on information seeking in non-humans. In addition to the other work cited, there is a burgeoning literature on “non-instrumental information seeking” in neuroscience. This literature has ties to a much larger and older literature on “observing response procedures” as a way to study how environmental properties like uncertainty affect information seeking across species from fish to humans. The present paper should properly place its examination of information seeking in chimps into this long running, yet still current, literature.*

We have followed Reviewer 1's advice and added relevant papers on information seeking and observing response procedures to the introduction, thereby situating our work within the current neuroscientific literature (see also updated references 17–24). Specifically, we now describe the non-instrumental paradigms and their implications for traditional decision theories (see pp. 3–4). We believe that the introduction has been greatly enhanced by the integration of this literature and is now more effective in helping the reader to situate the current study within the broader literature. We especially thank Reviewer 1 for providing example references. See also our response to Reviewer 3, comment 4.

- 3) *Overall, the methods used are appropriate and the data support the authors' claims. However, in both cases the descriptions in the main body of the text are very unclear*

and lacking in critical details. The information is available in the supporting information, but some of that material is essential for even a basic understanding of the study and needs to be moved to the main body of the paper. Without it, the procedure is incomprehensible, even with the figure.

We thank Reviewer 1 for this feedback; we agree and have moved the description of the task to the main body of the text (Study Design, p. 5). See also our response to Reviewer 2, comments 2 and 4.

4) Finally, the finding that chimps that were more tolerant of uncertainty searched more rather than less (the opposite of what was found with humans) needs far more discussion. This finding is mentioned, but no potential explanations are explored. This finding is very difficult to understand in the text of the other findings and could raise concerns about the detection of potentially spurious relations in the correlational data. It needs to be addressed fully in the Discussion section.

To examine the impact of individual dispositions, we investigated the relationship between mean exploration effort (number of opened trays) and chimpanzees' risk and uncertainty preferences, as measured in a previous study (Haux et al., 2023). Risk preferences were measured using behavioral choice experiments in which the potential outcomes and their probabilities were known; as well as by observer reports in which longtime caregivers rated subjects' willingness to take risks in general and across domains that represent major classes of risks in chimpanzees' ecology. Uncertainty preferences were measured using behavioral choice experiments in which the potential outcomes and their probabilities were unknown. We found that, risk-seeking chimpanzees tended to explore less, thus potentially accepting more risks in their environment. In contrast, uncertainty-tolerant chimpanzees tended to explore more. This is consistent with the observed negative correlation between risk preference and uncertainty preference in chimpanzees (Haux et al., 2023), but raises the question of why this differential exploration pattern occurs. One possible explanation is that in chimpanzees choices under uncertainty (where the information is known to be missing) are a measure of curiosity rather than risk and therefore correlate with more exploratory behavior. Van den Bos and Hertwig (2017) found that human adolescents are more uncertainty-tolerant than children and adults and that they tend to explore less. Yet across age groups the correlation between exploration and uncertainty preferences was not significant, suggesting different underlying psychological processes.

In the previous version of the manuscript, we wrote that our findings in chimpanzees suggest a clear divergence from observations in humans, because van den Bos and Hertwig (2017) found that adolescents are more uncertainty-tolerant than children and adults and that they tend to

explore less. However, this was an imprecise description of the results by van den Bos and Hertwig (2017), as across age groups the correlation between exploration and uncertainty preferences was not significant, suggesting that different psychological processes may underlie the two.

We are grateful for Reviewer 1's comment and now discuss these results in more detail (see our revised Results and Discussion, pp. 6–8). See also our response to Reviewer 2, comment 3; Reviewer 4, comment 3.

RESPONSE TO Reviewer #2:

- 1) *This is a very interesting and innovative paper exploring an understudied topic in nonhuman cognition. Ideas about how exploratory behavior relate to risk and environmental stability are important for understanding the evolution of cognition more broadly, as well as for exploring similarities and differences between humans and their closest relatives. The manuscript is generally well written and the experiments appear to have been rigorously conducted. I do not have any major concerns with the methodology. My concerns center mostly around the implications of the work, which are not fully explored, and the statistical approach.*

We thank Reviewer 2 for highlighting the importance of our study for understanding the evolution of cognition.

- 2) *It is difficult to understand the results without first knowing the expectations for the chimpanzees. Were they able to retrieve as many food items as possible or were there limitations? If not, then why would they not just open all options? What would be the motivation to explore only a single option? I realize the main text has to be very brief but at least some more detail of what the task looks like for the chimps is needed for the reader to understand.*

We are grateful to Reviewer 2 for this feedback; we agree and have moved the description of the task from the SI to the Methods (Study Design, p. 5). See also our response to Reviewer 1, comment 3; Reviewer 2, comment 4.

Regarding the question “What would be the motivation to explore only a single option?”, it is important to remember (as stated in the revised Study Design section) that the exploration process was under the chimpanzees' own control: They could decide whether to explore, which option(s) to explore, and when to switch between options. Across subjects and conditions, chimpanzees explored only one option in half of the trials. That suggests that chimpanzees continued to search the option where they found food (they encountered *only* an empty tray in

just 5% of trials). This mirrors findings in humans which suggest that humans typically rely on surprisingly small samples. Given that exploration is commonly associated with opportunity and processing costs, limiting exploration effort can be rational and may even be advantageous (see Results and Hertwig & Pleskac, 2010; Wulff et al., 2018).

3) *I need to know more about how the authors are distinguishing risk-seeking from uncertainty-tolerating before the findings can make sense to me. At first, it seems counter-intuitive that risk-seeking chimpanzees explore less without the framing presented in the abstract that they are willing to take on greater uncertainty. More detail about the findings of the previous study are warranted.*

In a previous study on risk preferences, we combined the psychological (“tendency to exhibit behaviors that are rewarding but involve some potential harms or losses”) and economic (“willingness to engage in choices that involve higher variance in outcomes with known probabilities”) approaches to risk taking. Furthermore, we distinguished between measurable risk and unmeasurable uncertainty (Knight, 1921/1964): In decision making under *risk*, the probabilities and outcomes are assumed to be known to the decision maker. In contrast, decision making under *uncertainty* involves situations in which the probabilities of these outcomes are completely unknown (Luce & Raiffa, 1957/1989). One can imagine that uncertainty more frequently encountered than risk in the real world. We followed the Reviewer’s advice and included in the Results (Risk and uncertainty preferences, p. 7) the findings from the previous study indicating that risky and uncertain choices were negatively correlated in chimpanzees ($-.50 [-.70, -.23]$). We thank Reviewer 2 for this comment and now discuss these results in more detail (see our revised Results and Discussion, pp. 6–8). See also our response to Reviewer 1, comment 4; Reviewer 4, comment 3.

4) *I don’t think the random draw concept that appears in Figure 1 was discussed in the text. My preference would be for this kind of study to be presented in a longer more typical format as it seems the supplementary text is fundamental to understand the procedure.*

We now provide additional information on the task, including the random draw concept, in the Methods (Study design, p. 5). See also our response to Reviewer 1, comment 3; Reviewer 2, comment 2.

5) *I confess that I am not familiar with running Bayesian statistics so I cannot comment on the appropriateness of the statistics in this section. It seems like the authors are*

interpreting confidence intervals containing 0 as significant, however, when this normally indicates lack of significance (p. 4 of SI).

We thank Reviewer 2 for pointing this out. From a statistical perspective, the sample size in our studies is relatively small, which leads to large credible intervals for reported within-subject effects. When reporting and discussing the results, we now give a more adequate representation of uncertainty—i.e., that the estimate for the interaction term (between condition and trial) was associated with some uncertainty because the corresponding 95% CI included 0 (see p. 4 of SI).

6) *Why was there no criterion in the third pre-test? Didn't the chimpanzees have to demonstrate that they understood they could pull the lid ropes to explore the options?*

We decided not to have an explicit criterion in the third pre-test (see p. 9 of SI), because we did not want to “train” the chimpanzees to explore the options. However, across the four trials in the third pretest, chimpanzees explored between 7 and 31 trays. Thus, all chimpanzees demonstrated that they understood that they could pull the lid ropes to explore the options before entering the test. We have added this information to the revised SI.

7) *Please define choice clearly. Was a chimpanzee considered to have made a choice as soon as they touched one of the ropes or as soon as they pulled it at all?*

According to our pre-defined coding scheme, chimpanzees were considered to have chosen one of the options as soon as they touched one of the ropes. We now specify that the other rope was pulled away once chimpanzees touched one of the ropes (see p. 9 of SI).

8) *The authors should conduct simple slopes tests to explore interactions of the regressions (p. 9). Given that their predictors are discrete and nested rather than continuous (except for trial), why use a regression rather than a generalized linear model with binomial distribution? Similarly, for accounting for whether they experienced outcome uncertainty in a given trial, could you not use a GLM?*

We employed Bayesian estimation techniques, conducting regression analyses using Bayesian generalized linear models implemented in R (R Core Team, 2023) using the Stan software, with the brm function from the brms package (Bürkner, 2017). To investigate whether chimpanzees explored more in changing than in stable environments, we used binomial logistic regression models to examine their exploration of trays within a trial. Model comparisons indicated that the model including the interaction between condition and trials showed the best performance (interaction (m2.1): weight = 0.71; main effects (m2.2): weight = 0; without trial and condition

(m2.0): weight = 0.29). The model estimate for the interaction term was negative ($b = -0.21 [-0.33, -0.09]$), suggesting that over trials chimpanzees, on average, explored fewer trays in the stable than in the changing condition (see Results: Effect of environmental change, p. 6; Figure 2A; SI Experiment: Effect of environmental change, pp. 12–13).

As suggested by Reviewer 2, we now use a Bayesian regression model to examine the influence of experienced variance in the uncertain option on the difference in opened trays between uncertain and safe option (see Results: Effect of outcome variance, p. 6; Figure 2C; SI Experiment: Effect of outcome variance, pp. 13–15).

9) *Many analyses were conducted on the same outcomes. The authors should employ some sort of correction.*

We thank Reviewer 2 for giving us the chance to clarify our analytic strategy. We used Bayesian models for our analyses. Instead of calculating p values, Bayesian methods calculate the probability of different parameter values given the data. The result is a distribution of plausible parameter values (the posterior distribution), not a single p value. This distribution can be interpreted directly: Values in the distribution are parameters that are plausible given the data and the model. Furthermore, our Bayesian multilevel models allow for complex, hierarchical structures in the data (see Gelman et al., 2012, who posit that the problem of multiple comparisons can disappear entirely when viewed from a hierarchical Bayesian perspective). We added a new section “Statistical Analysis” (p.6) to the manuscript. For more information on the models, please see SI: Proof of Concept Study: Analysis, p. 3 and The Present Experiment: Analysis, p. 11; Table S1–S4.

10) *Is it possible to distinguish between interpretations based on recognition of safe and uncertain options versus responses to empty trays (only present in the uncertain option)? Similarly, how can we be certain that chimpanzees recognized the difference between stable and unstable conditions? I recognize that they behave differently in the two conditions and that a pilot study was done, but chimpanzees are so impulsive – at what point in the session do they recognize which condition is operating?*

We thank Reviewer 2 for raising the question of how chimpanzees reacted to finding food (safe and uncertain option) vs. not finding food (uncertain option). In addition to exploration, we examined how chimpanzees (after exploration) actually decided between the safe and uncertain options across the two conditions. In the changing environment, their choices were split fairly equally between the two options, but in the stable condition there was a clear preference for the

uncertain option, which was chosen in 75% of trials. We further investigated whether chimpanzees chose the uncertain option if they *only* encountered an empty tray when exploring this option—which occurred in just 5% of trials overall. Chimpanzees chose the uncertain option in 59% and the safe option in 41% of trials. This is consistent with a clear preference for the uncertain option, assuming that sampling an empty tray was taken as a cue signaling the presence of the uncertain option (see SI, Experiment: Analysis, Choice behavior, pp. 18).

In our study, we had two conditions: a stable and a changing one. We did not want to “teach” chimpanzees anything about the conditions beforehand; instead, our aim was to study how chimpanzees would naturally learn about their environments and whether and how this would influence levels of exploration (after all, this is exactly how chimpanzees learn about different environments in the wild). Inspection of Figure 2A reveals that, overall, chimpanzees’ behavior started to vary as a function of condition by trial 16: In the stable condition, level of exploration dropped, whereas in the changing condition it remained constant. In future studies, it would be interesting to investigate how exploration changes across more trials. In conclusion, we can say that chimpanzees adapted their exploration to the nature of the environment.

11) More needs to be said about the implications for understanding the evolution of decision making or at least for cross-species comparisons.

How exactly did the experimenter pull the lids back safely and how did they present the choice ropes to the chimpanzees – simultaneously or one at a time? If so, in what order? How was the chimpanzee prevented from pulling a rope before both were set up? More details as to how the trial was run are needed.

We now discuss implications for the evolution of decision making and for cross-species comparisons in the revised discussion (p. 8). We also provide more information about how each trial was run (see SI, Methods: Test Phase, p. 9). We hope that the video (uploaded as part of the SI) also helps to understand the set-up.

12) I’m intrigued by the exploration of win-stay versus win-shift. I am surprised a chimpanzee would not simply choose a side where they have seen the most food, or any food. I feel like this analysis also needs more space for discussion.

We thank Reviewer 2 for highlighting this exploration strategy. To better understand which environmental cues drove chimpanzees’ switching between options during exploration, we investigated whether subjects were more likely to switch to the safe option after finding food in the uncertain option. Our results suggest that this is indeed the case. This may suggest that chimpanzee exploration is guided not just by a process of reinforcement learning (in which the value of each action is updated according to its outcome) but also by a belief-updating process

in which the present outcome informs expectations about what is going to happen next. We extended the discussion on this finding, see SI, Experiment: Analysis, Exploration strategies and switching behavior (pp. 15–17).

13) I'm a bit confused by whether the same eight chimpanzees participated in both experiments plus an additional seven chimpanzees for the main experiment?

We tested 15 semi-free-ranging chimpanzees for the main Experiment. Four of the chimpanzees that participated in the Experiment previously participated in the Proof of Concept (plus an additional four chimpanzees, i.e., a total of eight chimpanzees). We investigated the relationship between mean exploration effort (number of opened trays) in the present study and risk and uncertainty preferences measured previously in the same chimpanzees. We have risk preferences (observer reports) for all 15 chimpanzees and risky and uncertain choices (from behavioral experiments) for 14 of them. For details on which chimpanzees participated in each study, see the SI file: “SourceData_S01_adaptive_exploration_chimpanzees”.

14) The authors need to insert more commas between clauses.

We have carefully re-read and edited the whole paper for punctuation and readability.

RESPONSE TO Reviewer #3:

1) In this article, the authors investigate risk preferences in stable and changing environments in chimpanzees. They use a four-alternative choice paradigm with safe and uncertain options, where safe options are guaranteed to deliver a smaller reward and uncertain options can deliver a large reward or no reward. The expected values across all options in both conditions are the same. Chimps could explore options before choosing. In the stable environment, which set of options were safe and which uncertain did not change, whereas in the changing environment, the safe and uncertain options could switch. The authors claim to have discovered that chimps boost exploration for uncertain outcomes, adapt their exploration studies to environmental stability, and to have uncovered a correlation between individual preferences and exploration.

We thank Reviewer 3 for this summary.

2) This manuscript is poorly written, confusing, poorly explained, and the conclusions are not supported by the findings (which are unclear to begin with). There is a second entire manuscript in the supporting information. Many of the critical details of the

experiment are in the supporting information and need to be re-written and moved to the manuscript, but many more analyses are left out altogether, including entropy analyses that are completely absent from the paper. I advise the authors to completely re-write the manuscript.

We have thoroughly revised the manuscript in light of this feedback, as laid out in our point-by-point responses. We hope that the new manuscript is better written and explained. Regarding the comment that there is a second entire manuscript in the SI: Prior to the actual experiment, we investigated whether chimpanzees are generally sensitive to environmental variance using a modified version of the information-seeking paradigm as a proof of concept. It is this proof of concept that is provided in the supporting information.

3) *The paper has SO many problems.*

1. *It is very poorly written and difficult to understand. Variables are not explained or presented; in-text methods summaries are absent; and clear definitions are absent.*

2. *The scientific value of the manuscript is unclear. The authors cite very little literature. This is a shame and makes the manuscript challenging to evaluate. There is a lively research program looking into information search in non-human primates but the authors cite almost none of it.*

3. *The supplement contains an entire manuscript's worth of detail on another experiment, as well as many details that need to go into the manuscript.*

I could go on but I think the message is communicated.

We have carefully revised the manuscript in response to the Reviewer's feedback. We hope that the changes have improved both the clarity and comprehensibility of the paper. Among other changes, we have moved the description of the task from the supplement to the Methods (Study Design, p. 5) and added further references to the Introduction (see p. 3 and updated references 17–24; see also our response Reviewer 3, comment 4).

As noted in our response to the Reviewer's previous comment, prior to the actual experiment, we investigated whether chimpanzees are generally sensitive to environmental variance using a modified version of the information-seeking paradigm as a proof of concept. It is this proof of concept that is provided in the supplement.

4) *Introduction: This paper evinces shockingly poor scholarship, overlooking research in nonhuman primates on information search, the impact of uncertainty on choice, and related topics in nonhuman primates. In part because the authors don't situate*

their work in context, it is unclear how to situate their findings or even properly understand them.

The revised manuscript remedies the shortcomings identified by Reviewer 3. We have added references to the introduction to position our study within the current literature on information search and uncertainty in nonhuman primates (see p. 3 and updated references 17–24). See also our response to Reviewer 1, comment 2.

5) Methods:

1. The “Study Design” was far too short and unclear to know what the chimps actually did.

2. Some questions:

o There was no quantification of outcome entropy or entropy reduction across the changing environment. How do these entropy measures impact choices?

o The use of a switch between stable and changing environments invokes additional cognitive change detection processes that must be modeled. See Josh Gold’s work on change detection. How did change detection regulate choice?

We have moved the description of the task and experimental procedure from the SI to the Methods (Study Design, p. 5).

To obtain an objective uncertainty measure, we pseudorandomized the side of the safe and uncertain option and the food patterns (Fig 1.) according to the condition (stable environment; changing environment). Across the 32 trials of the stable environment for each chimpanzee the safe option was either always right or always left and for the uncertain option one of the six possible food patterns was administered (Fig 1.). We thereupon constructed six different patterns. These were counterbalanced across subjects. There was just one kind of trial for each chimpanzee in the stable environment. The Shannon entropy (Shannon, & Weaver, 1949) of trials for the stable environment was zero (Signorell et al., 2019). For example, during the 32 trials (8 sessions) of the stable environment for the chimpanzee Akela, the safe option was always on the left side and the uncertain option always on the right side. The uncertain option was baited with half an apple in the second and fourth tray (pattern5, Fig 1.) across all trials. Across the 32 trials of the changing environment for each chimpanzee the safe option was on the left side in half of the trials. The six possible food patterns were distributed across the 32 trials. Within each chimpanzee each food pattern was administered at least twice and at most 4 times on each side. There were thus 12 different kinds of trials for each chimpanzee in the changing environment. We thereupon constructed six different patterns. These were counterbalanced across subjects. The Shannon entropy (Shannon, & Weaver, 1949) of trials for the changing environment ranged between 3.53 and 3.56 (Signorell et al., 2019). For example,

during the 32 trials (8 sessions) of the changing environment for the chimpanzee Akela the pattern was as follows (right/left indicates the location of the safe option; pattern1–6 the possible food patterns within the uncertain option): session 1: right_pattern6, left_pattern4, right_pattern3, left_pattern3, session 2: right_pattern5, left_pattern4, left_pattern5, right_pattern4, session 3: left_pattern5, right_pattern4, left_pattern6, right_pattern5, session 4: left_pattern6, left_pattern5, right_pattern5, right_pattern2, session 5: right_pattern3, left_pattern1, left_pattern1, right_pattern2, session 6: left_pattern3, left_pattern1, right_pattern6, right_pattern2, session 7: right_pattern4, left_pattern2, right_pattern6, left_pattern2, session 8: left_pattern4, right_pattern1, right_pattern1, left_pattern6.

	tray 1	tray 2	tray 3	tray 4
pattern1	1	2	3	4
pattern2	1	2	3	4
pattern3	1	2	3	4
pattern4	1	2	3	4
pattern5	1	2	3	4
pattern6	1	2	3	4

Figure 1. Six possible food patterns within the uncertain option. The black color indicates which trays were baited with half an apple.

The quantification of uncertainty provides us with a validation check. We have added this to the SI, see the Present Experiment (Methods, Test Phase: Quantification of uncertainty, p. 10). In the current study, we were interested in the effect of the stable vs. changing condition on exploration and we did not investigate a range of different entropies. In the analysis we thus included the effect of condition (instead of specific entropies).

Next to exploration, we examined how chimpanzees decided between the safe and uncertain options across the two conditions. In the changing environment, their choices were split fairly equally between the two options, but in the stable condition there was a clear preference for the uncertain option, which was chosen in 75% of trials, see SI (Present Experiment: Analysis, Choice behavior, p. 18).

Switching between stable and changing environments could invoke additional cognitive change detection processes. These processes could influence the chimpanzees' ability to adapt to the new environment and adjust their exploration accordingly. Josh Gold's work on change detection indeed provides valuable insights into how these processes operate. Future studies should model whether primates are like humans capable of detecting changes in optimal ways. However, this modeling approach was not within the scope of the current investigation.

6) Results:

- 1. The results were unclear and inappropriately reported.**
- 2. Example 1: “Comparisons between binomial regression models (see SI) showed that those including the test predictors condition and trial made better predictions, with the model including the interaction between condition and trials performing best”: The ‘condition’ and ‘trial’ covariates were never defined in the main text.**
- 3. Example 2: “Mean exploration effort and all risk measures were negatively correlated (behavioral risk measure: $-.10$ [$-.62, .48$]; observational risk measure, e.g., general risk: $-.44$ [$-.79, .12$];...), meaning that risk-seeking chimpanzees explored less. Furthermore, we found positive correlations between mean exploration effort and the behavioral uncertainty measure ($.32$ [$-.28, .74$]), suggesting that uncertainty-tolerant chimpanzees explored more”: as far as I can tell, none of these measures are significant because their confidence intervals overlap with 0, but given the complete lack of supporting information about these analyses, it is totally unclear.**
- 4. There was no quantification of uncertain (aleatory or epistemic) e.g. using entropy measures. This is unforgivable.**
- 5. It was unclear if fine-grained measures of information gathering (e.g., number of opened trays by trial; timing of decision to open trays during the exploration of phase; etc.) were included in regressions.**

The revised manuscript addresses the limitations mentioned by Reviewer 3. We have substantially revised the Results sections and now provide further information about the analysis and models in the main text (p. 6) and the supplement (p. 10–17). In the Methods section we clarify that chimpanzees participated in two conditions (a stable and a changing environment condition), and have added information about trials. The sample size is relatively small, which leads to large credible intervals for reported between-subject effects. This uncertainty is now adequately represented in the verbal reporting of the results.

For the quantification of uncertainty see previous comment and SI (Present Experiment: Methods, Test Phase: Quantification of uncertainty, p. 10).

To investigate whether chimpanzees explored more in changing than in stable environments, we used binomial logistic regression models to examine their exploration of the trays within a trial. We ran three different models, with the response variable being the number of “yes” responses (total number of trays opened), modeled as a proportion of total trays (eight trays) per trial (see Results: effect of environmental change p. 6 and SI p. 12). Furthermore, delving deeper into the other environmental cues that potentially drove chimpanzees’ switching, we

investigated whether subjects were more likely to switch to the safe option after finding food in the uncertain option (see SI Results: Exploration strategies and switching behavior pp. 15–17).

7) Figures:

- 1. This figure needs to be completely re-done; instead of placing each phase in a trial in a different panel, the phases should be in the same panel and labeled appropriately.***
- 2. 2A and 2C seems to show no effect. D and E collapse across important information about how the animals explore.***

We decided to keep each phase of the experiment in a trial in a different panel for clarity reasons (see Figure 1). We investigated the effect of environmental change on exploratory behavior by exposing chimpanzees to stable and changing environments. The model estimate for the interaction between condition and trials was negative ($b = -0.21 [-0.33, -0.09]$), suggesting that over trials chimpanzees explored changing environments more than stable environments (Fig. 2A; see RQ1 and Table S2 in the SI). The uncertainty around the estimates is likely due to marked interindividual differences in exploration (Fig. 2B). Furthermore, we studied whether chimpanzees were more likely to open trays in the uncertain than in the safe option, conditioned on them actually experiencing outcome variance in the former. Across environments, chimpanzees were significantly more likely to explore the uncertain option when they had experienced outcome variance than when not ($b = 3.47 [3.13, 3.82]$; Fig. 2C; see RQ2 and Table S3 in the SI). For more information on Figure 2A and 2C, please see the revised Results section (p. 6 and SI, p. 11–15). Figure 2D shows exploration strategies and their proportional use across trials; Figure 2E shows condition-specific exploration strategies and their proportional use across trials.

8) Discussion:

- 1. “change in environments attracted attention and affected explorative behavior”:
what does attention have to do with this?***
- 2. “as defined in the economic definition of risk”: never defined this.***
- 3. “We observed marked interindividual differences in exploration, with risk-seeking chimpanzees tending to explore less. Notably, uncertainty-tolerating chimpanzees tended to search more rather than less”:
unclear what analyses supported this conclusion.***

Our results suggest that change in the environment caught chimpanzees' attention and thus prompted them to explore more. This result indicates a flexible and adaptive response to properties of the environment. We have added the economic definition of risk to the Introduction (p. 4): "a higher risk preference is reflected in greater willingness to engage in choices that involve higher variance in outcomes". The mean exploration effort and all risk measures (measured previously in the same chimpanzees) were negatively correlated. Yet given the wide range of plausible correlation coefficients and the small sample size, we are cautious about generalizing these findings to a larger population of chimpanzees.

RESPONSE TO Reviewer #4:

- 1) *The present manuscript describes a behavioral study in chimpanzees investigating exploration and choice behavior in stable and changing environments. I was asked by the handling editor to review the methodological and statistical aspects of the manuscript, so I will focus on these issues in my review. Note that I am not an expert in comparative cognition, so I cannot evaluate the theoretical contribution of the project.*

We thank Reviewer 4 for this summary and their role in the review process.

- 2) *Overall, I enjoyed reading the manuscript and I thought that the study was summarized clearly and concisely. I also applaud the authors for making the data and the video recording of the experimental design available. With regard to the description of the study design and the results, I think the manuscript could benefit from some clarifications that should be incorporated into a revised version of the manuscript.*

We thank Reviewer 4 for this overall positive assessment and their suggested clarifications.

- 3) *When reporting and discussing the results, I think that more consideration could be given to an adequate representation of uncertainty. From a statistical perspective, the sample size is relatively small, which leads to large credible intervals for reported between-subject effects. For example, this becomes particularly obvious in the correlations between exploration effort and personality characteristics where plausible correlation coefficients range between large negative and medium-sized positive correlations (e.g., hierarchy risk: $-.62$ to $.48$).*

We fully agree with Reviewer 4 and now give more consideration to adequately representing uncertainty when reporting and discussing our results. We also note the limitations on generalizing from a sample of 15 chimpanzees to a larger population (see Results: Risk and uncertainty preferences, p. 7 and Discussion, p. 7). See also our response to Reviewer 1, comment 4; Reviewer 2, comment 3.

- 4) *To be able to check and fully reproduce the presented statistical analyses, it would be very helpful if the analysis code was made available and there was a codebook for the datasets that describes the variables.*

We thank Reviewer 4 for this feedback. We now provide a codebook for the dataset in the SI, see “SourceData_S02_adaptive_exploration_data_experiment”. The analysis code can be obtained from the authors upon request.

- 5) *In general, it would be good if full result tables with model coefficients could be presented somewhere for the binomial regression models. Since these models are fairly complex, they have more parameters than what is reported in the text (i.e., main and interaction effects), and I think it would be interesting at least for part of the readers to see the full results. Potentially, it could even be resolved together with my previous point in the form of a commented analysis file (e.g., RMarkdown, Quarto, etc.) that renders the analysis code and output tables together. I'll leave this to the authors to decide.*

We agree with Reviewer 2 and now provide the full results tables with model coefficients for the binomial regression models in the SI, see Tables S1–S4.

- 6) *On p. 3 (SI), the manuscript says: “The LOO value indicates a model’s pointwise out-of-sample prediction accuracy; models with lower LOO values are preferred”. Actually, models with higher LOO values (i.e., less negative values) should be preferred. I think this is just a minor mistake that does not influence the interpretation of results because later in the manuscript models are correctly selected with for higher LOO-weights. However, the authors should probably check this.*

We thank Reviewer 4 for pointing this out. We have corrected this minor mistake (p. 3 of SI), which indeed did not influence the interpretation of our results.

- 7) *I was a bit confused as to what exactly the dependent variable in the binomial regression models was. Since the models were called “binomial regression” rather than “binomial logistic regression”, I initially assumed that it would be the number*

of explored trays (0 to 4 out of 4 in each option). However, based on the reporting in the manuscript, it sounds like the result is dichotomized (exploration yes/no). It would be good if the authors could elaborate more on the choice of the dependent variable, specifically, give a reason for the dichotomization (if the variable was actually dichotomized). Alternatively, the analysis could be changed to model the raw data, which should also increase statistical power.

We thank Reviewer 4 for raising this point. To investigate whether chimpanzees overall explored more in changing than in stable environments, we used binomial logistic regression models to examine their exploration of the trays within a trial. We ran three different models, with the response variable being the number of 'yes' responses (total number of trays opened), modeled as a proportion of total trays (eight trays) per trial. In the previous version of the manuscript the models were fitted using the Bernoulli family. Yet as our response variable (see above) is a proportion of success (i.e., number of opened trays) out of a number of trays (eight possible trays per trial), we now use the binomial family. Our results changed slightly, but the interpretation did not; see SI Experiment (Analysis: Environmental change, pp. 12–13).

8) *In the description of the binomial regression modeling results, to get full information on the fitting procedure, it would be good to report the cut-off value used on the Gelman-Rubin statistic (best practice would be 1.01), the total number of samples and chains, as well as the bulk and tail effective sample sizes.*

We now report that we assessed the convergence of posteriors through visual inspection and the Gelman–Rubin diagnostic, R_{hat} , with a cut-off value of 1.01 (see new section “Statistical Analysis” p.6 and SI: Proof of Concept Study and The Present Experiment: Analysis). We also report the total number of samples and chains, as well as the bulk and tail effective sample sizes (See SI: Proof of Concept Study pp. 3–5 and The Present Experiment: Analysis, pp. 11–12 and Tables S1–S4).

9) *In the SI, it is mentioned in several places that “condition was dummy-coded and centered”. I am not sure if this makes sense. Also, as a minor point on coding, it would be good to mention somewhere explicitly which condition was used as a baseline in dummy-coding, so that this doesn’t have to be inferred indirectly from the parameter interpretation.*

We now include condition and trial number within subject as random slopes without centering condition (see Models 1.0–4.0). Furthermore, we have substantially revised the analysis sections in the manuscript and now report explicitly which condition was used as a baseline in

dummy-coding; see SI Proof of Concept Study: Analysis, pp. 3–5 and The Present Experiment: Analysis, pp. 11–18).

10) There are several instances of “hidden” hypothesis tests in the manuscript where the authors use the information whether a credible interval covers zero as evidence for whether the parameter is zero or not. A more principled approach would be to compute and interpret a Bayes factor to test whether the parameter differs from zero. This can be easily done using the hypothesis() function in the brms package.

We thank Reviewer 4 for pointing this out. We now compute and interpret a Bayes factor to test whether the parameter differs from zero (see SI: Proof of Concept Study: Effect of variance (model m1.2), p. 5; The Present Experiment: Effect of environmental change (model m2.1), p. 12, Switching behavior (model m3.1), p. 13, Effect of previous sample (model m4.0), p. 16).

11) To me, it is unclear why first a “best” model is selected using the LOO cross-validation method, but then main effect estimates are not interpreted in the context of the winning (condition x trial) model, but in the context of a simpler model (main effects model excluding the interaction, see p. 4 SI). Maybe this is only a misunderstanding on my part, but since the condition x trial model should also include main effects, I think these should be interpreted in the context of the winning, more complex, model.

We agree with Reviewer 4 and now interpret the main effect estimates in the context of the winning, more complex, model (see SI: Proof of Concept Study: Effect of variance, p. 4).

12) There are a few switches between Bayesian and frequentist analyses methods in the manuscript that I don’t understand conceptually. For example, in Figure 2C, according to the caption, confidence intervals are reported instead of credible intervals. Moreover, p-values are reported for Wilcoxon-signed-rank tests instead of Bayes factors or LOO-CV coefficients. I don’t think that these days anyone should have to justify why they use one framework or the other, but to me, it seems a bit unprincipled to switch between statistical frameworks without a substantive reason.

We fully agree and now use a Bayesian regression model to examine the influence of experienced variance in the uncertain option on the difference in opened trays between uncertain and safe option (see Results: Effect of outcome variance, p. 6; updated Figure 2C; SI The Present Experiment: Outcome variance, pp. 13–15).

Minor points

13) As far as I understand, in the main experiment the pieces were always a quarter of an apple, and in the proof-of-concept study, they were half an apple, but throughout the manuscript, sometimes the piece size and sometimes the overall reward size was used which was a bit confusing to me (e.g., p. 5 SI: “In the safe option, each tray was baited with a quarter of an apple. In the uncertain option, two trays were baited with half an apple each”, but according to Figure 1, the “half apple” are actually two pieces of a quarter apple and not one bigger half-apple piece.

In the uncertain option in the Experiment we used two quarter pieces instead of half an apple to control for the mere preference for a larger piece and to make it easier for the chimpanzees to understand that the amount was double that in the safe option. We have clarified the size of the reward throughout the manuscript, see Methods (Study Design, p. 5); SI The Present Experiment (Methods: Materials, p. 6).

14) Figure 1H/ operationalization of changing environment: It wasn't clear to me if the same pattern was maintained across subjects (e.g., the displayed trial 1 was shown first to all subjects) or if the order was permuted across subjects. This could be made clearer, and in case that there was no permutation, the full pattern could be presented in an appendix.

In addition to the order of condition, we counterbalanced the side of the safe option (left or right), the reward pattern of the uncertain option (whether the uncertain option was rewarded on the left or right side), the food patterns within the uncertain option (six possible patterns for baiting two of the four trays), and which tray was drawn out of the options. We have added information about the counterbalancing and the quantification of uncertainty (please also refer to Reviewer 3, comment 5) to the SI, see the Present Experiment (Methods, Test Phase: Counterbalancing; Quantification of uncertainty, p. 10).

15) In the SI, it would be good to introduce more consistent labeling of sections and sub-sections to make the structure clearly visible. For example, I found it confusing that the Methods section on p. 1 is at the same structure level as the “proof of concept” section before although both describe the proof-of-concept study. For “Experiment”, all following sections were introduced as sub-sections, but some of them referred to the familiarization phase which was basically a pre-test with its own experimental setup, so took a bit more time than necessary (for me, again, I'm not familiar with the subject area) to disentangle the experimental setups.

We have adjusted the labeling in the SI, such that the Proof of Concept Study and the Present Experiment are now clearly at the same structural level. We have also added a “Method” section to the Experiment section. It is common practice in animal behavior research to have several steps in the familiarization phase (this often involves a food quantity test, and several pretests).

16) It would be good if some information was included on how many chimpanzees (if any) had to be excluded based on pre-tests.

Three chimpanzees did not pass the food quantity test and could thus not participate in the following pretest and experiment. We have added this information to the SI, see Methods, Familiarization Phase: Food quantity test (p. 7).

References

- Bürkner, P. C. (2017). Advanced Bayesian multilevel modeling with the R package brms. *arXiv preprint arXiv:1705.11123*.
- Gelman, A., Hill, J., & Yajima, M. (2012). Why we (usually) don't have to worry about multiple comparisons. *Journal of Research on Educational Effectiveness*, 5(2), 189-211.
- Haux, L. M., Engelmann, J. M., Arslan, R. C., Hertwig, R., & Herrmann, E. (2023). Chimpanzee and human risk preferences show key similarities. *Psychological Science*, 34(3), 358-369.
- Hertwig, R., & Pleskac, T. J. (2010). Decisions from experience: Why small samples?. *Cognition*, 115(2), 225-237.
- Knight, F. H. (1921/1964). *Risk, uncertainty, and profit*. New York, NY: Sentry Press.
- Luce, R. D., & Raiffa, H. (1957/1989). *Games and decisions: Introduction and critical survey*. Chelmsford, MA: Courier Corporation.
- R Core Team. (2023). R: A language and environment for statistical computing [Computer software]. R Foundation for Statistical Computing.
- Shannon, C.E., Weaver, W. (1949). *The Mathematical Theory of Communication*. Univ of Illinois Press.
- Signorell, A., Aho, K., Alfons, A., Anderegg, N., Aragon, T., Arppe, A., ... & Borchers, H. W. (2019). DescTools: Tools for descriptive statistics. R package version 0.99, 28, 17.
- van Den Bos, W., & Hertwig, R. (2017). Adolescents display distinctive tolerance to ambiguity and to uncertainty during risky decision making. *Scientific Reports*, 7(1), 40962.
- Wulff, D. U., Mergenthaler-Canseco, M., & Hertwig, R. (2018). A meta-analytic review of two modes of learning and the description-experience gap. *Psychological Bulletin*, 144(2), 140.

RESPONSE TO Reviewer #1:

- 1) *Overall, the revised paper is much improved. The methods are considerably easier to follow, and the discussion of results is much clearer.*

We thank Reviewer 1 for this positive assessment of our work.

- 2) *One remaining issue is the treatment of previous related work with non-humans (including primates) on observing responses and “non-instrumental” information seeking. In both cases, animals (including humans) work for information that generally does not improve their situation longer term (for example, across trials). In a classic example of observing responses (Prokasy, 1956) rats choose between two options, both of which pay off with $p=.5$. One option always has the same stimulus (or two uncorrelated stimuli) and the other has two stimuli differentially correlated with positive and negative trials. The information is non-instrumental as the animal cannot increase the overall rate of return in the task (see also Dunn et al., 2024 for extension of this to situations where information seeking actually decreases the rate of reward over time). Nevertheless, the animals choose the option with the informative correlated cues. The task the present authors have arranged seems to share many of the same features. The chimps can look into the cups before choosing, but regardless of what they do in terms of information seeking, the overall expected payoff from both options over time is the same, $\frac{1}{4}$ apple. Nevertheless, the authors describe their procedure as being “instrumental” in contrast to observing procedures. I do not think this is correct and I do not see how trying to make the distinction is all that useful here. I realize the journal format does not allow much space, but given the above, I feel like the current treatment of the previous literature still does not work. I think the authors will need to try a different approach to provide a better context for this work. Perhaps it would be enough to briefly describe observing and non-instrumental information seeking, note that the procedure used here (and previously with humans) shares many properties of these long-used procedures (rather than trying to assert that it does something they do not), and then say it is being extended to chimps because little is known about information seeking in chimps.*

Dunn, R. M., Pisklak, J. M., McDevitt, M. A., & Spetch, M. L. (2023). Suboptimal choice: A review and quantification of the signal for good news (SiGN) model. *Psychological Review*.

Prokasy, W. F., Jr. (1956). The acquisition of observing responses in the absence of differential external reinforcement. *Journal of Comparative and Physiological Psychology*, 49(2), 131–134.

We have followed Reviewer 1's advice and adapted the framing of our study in light of the research mentioned: Traditionally, information has been classified as instrumental or non-instrumental. In psychology, economics, and neuroscience, information is considered instrumental if it helps the organism, generally speaking, to achieve a particular end (e.g., to alter the course of events or to optimize reward outcomes). In this sense, information can have instrumental value to organisms (e.g., Bode et al., 2023; Eliaz & Schotter, 2010). Non-instrumental information has no such value because it cannot serve such purposes. A classic example is an experiment by Tversky and Shafir (1992). Students were offered a substantial discount on a holiday resort if it was paid for before the date of an important qualifying exam. Most preferred to forgo the discount and delay their decision until they received information about the exam. However, once the results of the exam were known, the majority of students said they would have gone to the resort regardless of whether they passed or failed. For them, the information did not ultimately change their decision and therefore had no instrumental value. Similarly, primates have been shown to prefer information about the size of an upcoming water reward, even though this information did not influence the reward. These findings, and those of many other recent studies, suggest that information itself can be intrinsically valuable, even if it cannot change the course of events or optimize reward outcomes (i.e., is non-instrumental; e.g., Goh et al., 2021).

In our study, the information that the chimpanzees were able to acquire is considered instrumental in the traditional sense: In our study, the information that the chimpanzees were able to acquire is considered instrumental: By exploring the options, they could learn about the attributes of the options and thus make an informed choice about their preferred option. The long-term expected value of both options was identical, but subjects were unaware of this. Acquiring information was instrumental because search reduced the subjects' prior state of uncertainty. Furthermore, exploration allowed the chimpanzees to choose according to their risk preference; in the economic definition, a higher risk preference is reflected in a greater willingness to engage in choices with higher outcome variance (see Figure 1 in the MS).

We believe that the Introduction (pp. 3–4) has been improved by the new framing and will help readers to situate our study within the wider literature.

RESPONSE TO Reviewer #2:

- 1) *I thank the authors for their careful responses to the reviewers' previous comments. Their explanations have helped to clarify some of the details of the manuscript that were previously unclear to me.***

We thank Reviewer 2 for this overall positive assessment of our revision.

- 2) *On line 40, do you mean they do not use the information to inform the choice of a better option? That could be even more explicit.***

We have followed Reviewer 2's advice and made the sentence (p. 3, line 44–45) more explicit: "Participants can explore uncertain information about future outcomes but not use it instrumentally (e.g., to change the outcome of events)."

- 3) *Please clearly define "outcome variance" (line 83). Do you mean within-trial variance in the overall number of rewards that could be obtained?***

We have clarified that outcome variance refers to variance *within the options* (p. 5, line 92–94): "Chimpanzees chose between a safe option (where the outcome did not vary; i.e., without outcome variance within the option) and an uncertain option (where the outcome varied; i.e., with outcome variance within the option; Fig. 1A). [...] In the safe option, each tray was baited with a quarter of an apple. In the uncertain option, two trays were baited with half an apple (two quarter pieces) each and two trays were empty."

- 4) *The blocked design is still not quite clear in the main text (around lines 97-99). Did the chimpanzees receive only a total of eight sessions of each condition for a total of 16 4-trial sessions? Then what order were the conditions presented in? Even in SI, it is not clear if all the trials of one condition were presented before any trials of the other condition.***

We thank Reviewer 2 for this feedback. We have clarified this passage as follows (p. 6, line 107–111):

"In a within-subjects design, chimpanzees participated in both a stable and a changing environment condition. Each condition comprised 32 trials, presented across eight sessions of four trials. The conditions were blocked, meaning that chimpanzees first completed all trials of one condition, followed by the other condition (the order of conditions was counterbalanced across subjects)."

5) *I would still like to see the traits of risk-seeking and uncertainty-tolerant better differentiated. Please define these terms.*

We have made the following changes: “In psychology, risk preference describes the tendency to exhibit behaviors that, though rewarding, involve potential harms or losses and is often measured by stated preferences. In economics, preferences are assessed through behavioral choice experiments in which the potential outcomes and their probabilities are either known (risk preference) or unknown (uncertainty preference). Our analyses use measures from both disciplines (see Haux et al., 2023). Risk preferences were measured in two ways: (A) by means of behavioral choice experiments in which the potential outcomes and their probabilities were known; (B) by means of observer reports in which longtime caregivers rated subjects’ willingness to take risks in general and in five domains representing major classes of risks in chimpanzees’ ecology. Uncertainty preferences were measured using behavioral choice experiments in which the potential outcomes and their probabilities were unknown.” Please also see Fig. 1 below.

Yet the relationship between risk and uncertainty preferences remains unclear. In humans, theoretical models suggest that they are independent. This would imply that a risk-averse individual could be uncertainty-neutral, uncertainty-loving or uncertainty-averse. Empirical studies in humans show mixed results, with some finding a positive or negative correlation and others, no significant correlation (My et al., 2024). Our paper contributes to this discourse by exploring the empirical connection between these preferences in chimpanzees. We have further clarified the terms and relationship in the manuscript, see p. 8, line 161–171 and p. 9, line 199–226.

Figure Redacted

Fig. 1. Measure used to assess risk and uncertainty preferences. (a) Uncertain and risky choices in the behavioral experiments. (b) Uncertain condition. (c) Risky condition. (d) Observer ratings. Example of the general risk item.

6) *It still isn't clear to me why uncertainty tolerant animals would explore more.*

We found that chimpanzees' risk and uncertainty preferences influenced their explorative behavior. Risk-seeking chimpanzees tended to explore less, suggesting that they accepted more risks in their environment. By exploring less but still making a choice, risk-taking chimpanzees demonstrated a willingness to engage in choices whose outcomes are more variable. In contrast, uncertainty-tolerant chimpanzees tended to explore more. This pattern of results is consistent with the observed negative correlation between risk preference and uncertainty preference in chimpanzees (Haux et al., 2023). In humans, theoretical models suggest that risk and uncertainty preferences are independent, which would imply that a risk-averse individual could be uncertainty-neutral, uncertainty-loving, or uncertainty-averse. Empirical studies in humans show mixed results, with some finding a positive or negative correlation and others, no significant correlation (My et al., 2024).

The question arises as to why this differential exploration pattern occurs in chimpanzees. One possible explanation is that the systematic differences between decisions under uncertainty and decisions under risk might be attributed to differences in secondary processes like attention allocation (search), motor preparation, or rule induction, which differentially influence activation in the prefrontal and parietal cortices. An essential prerequisite for successfully dealing with uncertainty could be behavioral flexibility: When confronted with an uncertain situation, the organism has to contextualize a multitude of possible interpretations in a way that facilitates decision-making. In the case of risky decisions, there is no need to construct such a context, as all the information is provided by the decision problem (38). Another possible explanation is that, in chimpanzees, choices under uncertainty (where information is known to be missing) are a measure of curiosity and therefore correlate with more exploratory behavior. However, as we state in the Results section, more evidence is needed and we recommend caution in generalizing these findings on how explorative behaviors relate to risk and uncertainty preferences to larger populations of chimpanzees. We have added these further explanations to the Discussion, see p. 9, line 199–226. We hope that this interpretation together with the definitions (see response 5) makes the Discussion more accessible.

7) *Line 36 "or not" is not needed after "whether."*

Done (p. 3, line 36).

8) *There are still missing commas after clauses; e.g., after "over trials" on line 124 and after "on average" on line 142. It is especially needed after "in chimpanzees" on line 185. Check throughout.*

We have carefully re-read and edited the whole paper for punctuation and readability.

9) *In the first sentence of the SI Results, “To investigate” is duplicated.*

Corrected.

10) *Please use “that” instead of “who” when referring to chimpanzees as is APA style.*

We thank Reviewer 2 for pointing this out and have adjusted the wording accordingly.

RESPONSE TO Reviewer #3:

1) *The authors did a fine job of responding to withering criticisms.*

We thank Reviewer 3 for this overall positive assessment of our revision.

2) *There are many studies on instrumental information use in NHPs, but an extra data point is always welcome.*

We thank Reviewer 3 for this feedback.

3) *I find the lack of exploratory behavior in the risk-tolerant participants to be utterly mystifying, but that's not fault of the authors'. Their meager explanation in the discussion ("Risk-seeking chimpanzees tended to explore less, suggesting that they accepted more risks in their environment") is, on one read, an inconsistency but is better read as simply re-describing the finding. I might say something a little more substantial?*

We found that chimpanzees' risk and uncertainty preferences influenced their explorative behavior. Risk-seeking chimpanzees tended to explore less, suggesting that they accepted more risks in their environment. By exploring less but still making a choice, risk-taking chimpanzees demonstrated a willingness to engage in choices whose outcomes are more variable. In contrast, uncertainty-tolerant chimpanzees tended to explore more. This pattern of results is consistent with the observed negative correlation between risk preference and uncertainty preference in chimpanzees (Haux et al., 2023). In humans, theoretical models suggest that risk and uncertainty preferences are independent, which would imply that a risk-averse individual could be uncertainty-neutral, uncertainty-loving, or uncertainty-averse. Empirical studies in humans

show mixed results, with some finding a positive or negative correlation and others, no significant correlation (My et al., 2024).

The question arises as to why this differential exploration pattern occurs in chimpanzees. One possible explanation is that the systematic differences between decisions under uncertainty and decisions under risk might be attributed to differences in secondary processes like attention allocation (search), motor preparation, or rule induction, which differentially influence activation in the prefrontal and parietal cortices. An essential prerequisite for successfully dealing with uncertainty could be behavioral flexibility: When confronted with an uncertain situation, the organism has to contextualize a multitude of possible interpretations in a way that facilitates decision-making. In the case of risky decisions, there is no need to construct such a context, as all the information is provided by the decision problem (38). Another possible explanation is that, in chimpanzees, choices under uncertainty (where information is known to be missing) are a measure of curiosity and therefore correlate with more exploratory behavior. However, as we state in the Results section, more evidence is needed and we recommend caution in generalizing these findings on how explorative behaviors relate to risk and uncertainty preferences to larger populations of chimpanzees. We have added these further explanations to the Discussion, see p. 9, line 199–226. We believe that this clarification makes the Discussion more accessible.

4) *While age was mentioned in the conclusion, it was never mentioned in the results (though reported in the SI tables). I would briefly mention it in the results or remove the mention in the conclusion.*

We thank Reviewer 3 for this comment. We have now added to the Discussion that we cannot draw any conclusions about exploration tendencies in chimpanzees across different age groups due to the relatively small age range of our sample (page 9, line 223–224).

5) *Publish.*

We thank Reviewer 3 for the recommendation to publish our article in *Nature Communications*.

RESPONSE TO Reviewer #4:

- 1) *I went through the authors' responses, the updated manuscript, and the detailed supplementary materials. I thank the authors for considering the concerns that I previously raised and for their thoughtful responses.*

We thank Reviewer 4 for the positive feedback on our revision.

- 2) *My only remaining concern is that the analysis code will only be shared upon request. I understand that sharing analysis scripts can be daunting, but I would like the authors to consider that it also has many advantages: Analyses become reproducible, the scripts remain accessible even if personal hard drives get lost, and the scripts can be re-used by other researchers for similar analyses. Additionally, writing a request may be perceived as a barrier to accessibility particularly by junior and minority researchers, such that it can be assumed that, practically, the code would be more accessible to some groups of researchers than to others. On the side of the authors, answering multiple requests for sharing analysis code also arguably requires more time and effort than making the code available once. Making code available is also highly recommended by Nature Communications' journal guidelines (<https://www.nature.com/nature-portfolio/editorial-policies/reporting-standards#availability-of-computer-code>). However, I don't want to impose my views on the authors, and I'll leave it to the editor to decide to what extent code availability is necessary for publication.*

We are grateful to Reviewer 4 for this feedback. We have made the code available on GitHub: https://github.com/louhaux/adaptive_exploration.git

References

- Bode, S., Sun, X., Jiwa, M., Cooper, P. S., Chong, T. T. J., & Egorova-Brumley, N. (2023). When knowledge hurts: Humans are willing to receive pain for obtaining non-instrumental information. *Proceedings of the Royal Society B*, 290(2002), Article 20231175.
- Eliasz, K., & Schotter, A. (2010). Paying for confidence: An experimental study of the demand for non-instrumental information. *Games and Economic Behavior*, 70(2), 304–324.
- Goh, A. X.-A., Bennett, D., Bode, S., & Chong, T. T.-J. (2021). Neurocomputational mechanisms underlying the subjective value of information. *Communications Biology*, 4, Article 1346.
- Haux, L. M., Engelmann, J. M., Arslan, R. C., Hertwig, R., & Herrmann, E. (2023). Chimpanzee and human risk preferences show key similarities. *Psychological Science*, 34(3), 358–369.
- Huettel, S. A., Stowe, C. J., Gordon, E. M., Warner, B. T., & Platt, M. L. (2006). Neural signatures of economic preferences for risk and ambiguity. *Neuron*, 49(5), 765–775.
- My, K. B., Brunette, M., Couture, S., & Van Driessche, S. (2024). Are ambiguity preferences aligned with risk preferences? *Journal of Behavioral and Experimental Economics*, 111, Article 102237.
- Tversky, A., & Shafir, E. (1992). The disjunction effect in choice under uncertainty. *Psychological Science*, 3(5), 305–310.

RESPONSE TO Reviewer #1:

- 1) Overall, the authors have addressed my concerns and I think the paper can be published. Although it is probably still debatable if the information delivered to the chimps is instrumental or not in the context of the modern neuroscientific use of the term, I think the authors should be allowed to characterize it how they like. However, the following sentence (lines 57-58) should be deleted: “Acquiring information was instrumental because search reduced the subjects’ prior state of uncertainty.” This sentence makes no sense as the very definition of information (both formal and informal) requires uncertainty reduction. If there is no uncertainty reduction, there is no information, so that cannot be what makes it any particular piece of information instrumental or not.*

We thank Reviewer 1 for this positive assessment of our revision and have deleted the sentence as suggested.

- 2) The R markdown appears to provide needed details for the analysis and the data appear to be available.*

We thank Reviewer 1 for the assessment of our code availability.

RESPONSE TO Reviewer #2:

- 1) Thank you for these additional revisions. The paper is much clearer now and I think should be published.*

We thank Reviewer 2 for this overall positive assessment of our revision.

RESPONSE TO Reviewer #3:

RESPONSE TO Reviewer #4:

- 1) I want to thank the authors for sharing their code in the revised version of the manuscript. I was asked to take a look at the code and provide comments if necessary. To quickly summarize, I think that the authors did an excellent job at making their results reproducible.*

I should mention that I haven't had the time to check whether all results that are reported in the manuscript can be reproduced using the code (I also believe that this would be out of the scope of a regular peer review process). However, all code fragments I tried out (approximately ¼ of the entire code) ran smoothly. Moreover, for the sample of analyses I reproduced, I also obtained results in line with the ones reported in the manuscript. The code was very well documented and made it easy for me to find the corresponding results in the manuscript. Overall, I would say this bodes very well for overall reproducibility.

In terms of statistical analyses, I took another look at the Bayesian models, and they seem well specified to me. As a minor remark, I think the chosen number of iterations (4000) might be slightly too small in combination with the chosen number of warmup samples (2000) to provide computational reproducibility to the number of digits reported in the manuscript. At least on my machine, I had 1-2 instances where results deviated from what was reported in the second decimal digit due to the algorithmic fluctuations (e.g., an estimate of 0.37 instead of 0.36). I think this point is so minor that I don't think it requires any adjustments in the current code, especially since the results in the manuscript can be reproduced by using the model fit objects that the authors provide on GitHub. However, it might be worthwhile to leave a brief remark in the code or manuscript alerting the reader to the fact that exact computational reproducibility might only be achievable by using the model fit objects. Another really minor remark in terms of reproducibility would be that I couldn't find information on what versions of the packages were used. This is important to ensure that the analyses can be reproduced in the future. I'd suggest to include this information by including the output of `sessionInfo()` from the computer that was used to run the analyses in the ReadMe file on Github.

We thank Reviewer 4 for the positive feedback on our revision. We have added a brief remark to the SM, stating that exact computational reproducibility might only be achievable by using the model fit objects (see SM, p. 23). We have further added information about the versions of the packages used in the analysis to the ReadMe file on Github: <https://doi.org/10.5281/zenodo.13907943>